# WHAT DEEP NETWORKS WANT TO LEARN AND HOW TO GET THERE

## ABSTRACT

*Are we there yet?* It's hard to say when we don't know where we're going. The only thing that seems to stay the same in the rapidly evolving field of deep learning are long training runs until test performance saturates. Moreover, it's not clear when to stop training; practitioners observe extrinsic metrics like the training error, test error, and regularization terms but still might be tempted to stop early lest the deep network (DN) overfit. In this paper, we develop the first intrinsic, analytical, and interpretable characterization of where the training process is headed. The key is to analyze the geometry of the tessellation of the DN input space that is induced by a continuous piecewise-affine approximation to its input-output mapping. Analogous to the Voronoi tiling that underlies K-means clustering, each tile in a DN's power diagram tiling is parameterized by a centroid vector that equals the sum of the rows of the Jacobian of the DN input-output mapping. Our key result on learning is that a DN first reaches the point of generalization when the training data become aligned (in the sense of maximum cosine similarity) with the centroids of the tiles containing them. The DN then later reaches the point of maximum robustness when the training data become aligned with each row of the (rank-one) Jacobian. Hence, centroid and Jacobian alignment are the destination that learning algorithms aspire to reach. We leverage this new understanding in GrokAlign, a regularisation strategy for DN learning that provably and efficiently induces centroid and Jacobian alignment. Our experiments with convnets and transformers demonstrate that GrokAlign significantly accelerates delayed generalization (so-called "grokking") and improves robustness.

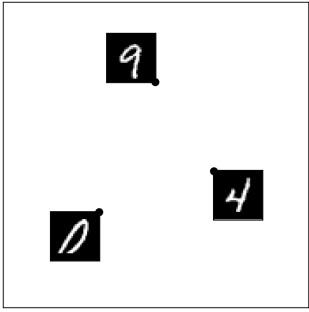 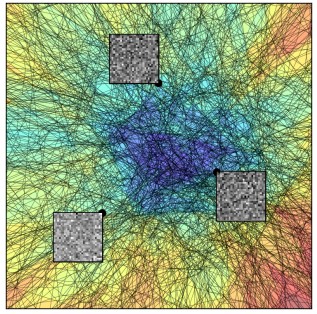 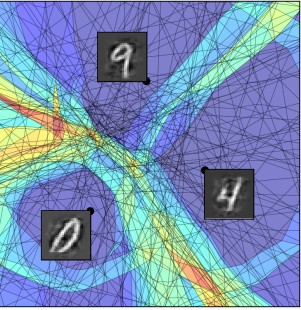

Training data points      Memorisation w/o generalisation      Robust generalisation

Figure 1: *Jacobian alignment* is the destination of training for deep networks (DNs) exhibiting *generalisation* and *robustness*. We study the simplification of the input-space partition of a DN induced by its continuous piecewise affine approximation. Alignment of the centroids of the partition regions to their corresponding training data points characterises the generalisation of a DN; the refinement of this property to Jacobian alignment characterises its robustness. We visualize a DN trained on MNIST along the 2D slice of its input space formed by three training samples (**left**). We plot the input-space partition plus centroids of the three regions containing the training samples at the points where the DN has memorised the training data without generalising (**centre**) and the DN has achieved Jacobian alignment and exhibits both generalisation and robustness (**right**).

# 1 INTRODUCTION

It is hard to know when to stop deep network (DN) training when we do not know where it is headed. Typically, practitioners stop when the improvement of DN extrinsic factors like training performance over training steps is minimal. This is done with the hope that the DN does not over-fit on the training data and exhibits *generalisation* – the ability to extrapolate its learning to unseen inputs. However, recent observations show that transformers trained on algorithmic tasks (Power et al., 2022) and fully connected DNs with large-norm initialization trained for image-classification (Liu et al., 2022) can exhibit *grokking* – the delayed onset of generalisation – long after the DN has reached zero training error (i.e., memorisation). The grokking phenomenon has prompted substantial research into explaining why delayed generalization happens (Nanda et al., 2022; Yunis et al., 2024a) and raises the ultimate question – *where are the training dynamics of DNs headed, such that generalization may occur long after over-fitting?*

Recently, Humayun et al. (2024) has observed an emergent robustness phenomenon with long training, for convolutional DNs performing image classification and transformers performing next-token prediction. In these cases, as training progresses for one or two orders of magnitude more epochs beyond generalisation, robustness seems to emerge naturally. This finding is at odds with the prevailing notion that robustness requires a separate form of training (i.e., adversarial training) to be achieved in conjunction with generalisation (Tsipras et al., 2019; Zhang et al., 2019). The emergent robustness phenomenon raises a question similar in tone to that on grokking – *where are the training dynamics of DNs headed, such that robustness may occur long after generalisation?*

In this paper, we argue that the destination where a DNs training dynamics are headed, is a state of *Jacobian-alignment* – a setting where the rows of the input-output Jacobian of a DN at any data point $\mathbf{x}$, become a scalar multiple of $\mathbf{x}$. Our argument is based on a careful study of the geometry of a DN (i.e, the arrangement of the input space partition induced by its continuous piecewise affine approximation), a simplification of which is related to the generalisation and robustness properties of a DN (Humayun et al., 2024). To explain this simplification, we propose using *centroids* as a summary of the DN's local geometry, where a *centroid* is defined as the sum of the rows of the DN's input-output Jacobian at a given input point (Balestriero et al., 2019). In Figure 1 we present how the centroids evolve as Jacobian-alignment is approached.

We show that DNs first exhibit *centroid alignment* – maximal cosine similarity between an input $\mathbf{x}$ and the centroid of the input-space partition region containing $\mathbf{x}$ – while achieving generalisation and refine the geometry of the DN to additionally achieve *Jacobian-alignment*, therefore exhibiting robustness. In the regime of DNs with rank one Jacobians, we prove that Jacobian alignment corresponds to the destination of 'optimal robustness' in DN training. When the Jacobian is not necessarily rank one, we leverage empirical simulations and the known biases of gradient descent to draw a similar conclusion. Our analytic characterisation provides a clear perspective of where the learning process of a DN is headed in a computationally accessible and compact form.

Apart from uncovering what the destination is, we also provide a method to reach the destination in practice and often in an expedited manner. Our theoretical framework inspires *GrokAlign* as a direct regularisation strategy to induce the alignment of DNs in practice. Our contributions are as follows:

- **[C1].** We formally characterise the simplification of a DN's geometry leading to maximal generalisation as the emergence of centroid alignment (see Theorem 4).
- **[C2].** We formally characterise the refinement of centroid alignment to Jacobian alignment that leads to optimal robustness for DNs with rank one Jacobians (see Theorem 7).
- **[C3].** Through numerous experiments, we demonstrate that the training dynamics of DNs seek Jacobian alignment, with standard training pipelines only weakly inducing alignment due to their inability to control the norms of the Jacobians (see Section 3.1).
- **[C4].** We introduce GrokAlign, a form of Jacobian regularisation, as a training strategy that directly induces Jacobian alignment (see Theorem 8).

Impacts of our analytic characterisation of where the training dynamics of DNs are headed include: (1) GrokAlign accelerates grokking more effectively than other regularisation strategies, including weight decay, Grokfast (Lee et al., 2024), and OrthoGrad (Prieto et al., 2025). (2) GrokAlign improves the alignment of a DN and thereby its robustness. (3) Centroid alignment is an efficient and intuitive metric for interpreting DN training dynamics and preventing under- or over-training.

## 2    THE DESTINATION OF DEEP NETWORK TRAINING

In this section, we show that Jacobian alignment characterises what DNs *should* want to learn. First, we introduce the functional geometry and neural tangent kernel of a DN. Second, starting from the premise that it is a simplification of the functional geometry of the DN that demarcates the destination, we illustrate how centroid alignment corresponds to generalisation in Section 2.2. Then we refine the centroid alignment property of DNs to Jacobian alignment to illustrate the robustness of this configuration in Section 2.3.

### 2.1    THE FUNCTIONAL GEOMETRY AND NEURAL TANGENT KERNELS OF DEEP NETWORKS

**Deep Networks.**    A DN $f : \mathbb{R}^d \to \mathbb{R}^C$ is constructed through a composition of layers which constitute affine transformations followed by a nonlinearity. For the task of classification, the DN predicts the class of a point $\mathbf{x}$ with $\mathrm{argmax}(f(\mathbf{x}))$. The DN is trained on a data set $\{(\mathbf{x}_p, y_p)\}_{p=1}^m$ – where $\mathbf{x}_p \in \mathbb{R}^d$ and $y_p \in \mathbb{R}$ is its corresponding class – under some loss function $\mathcal{L}_{\mathrm{Class}} = \frac{1}{m} \sum_{p=1}^m \ell(f(\mathbf{x}_p), y_p)$. Let $\mathbf{J}_\mathbf{x} \in \mathbb{R}^{C \times d}$ denote the input-output Jacobian of $f$ at $\mathbf{x}$, with the $c^{\mathrm{th}}$ row denoted as $\mathbf{j}_\mathbf{x}^{\cdot(c)}$. The output of the DN can then be decomposed as $f(\mathbf{x}) = \mathbf{J}_\mathbf{x} \mathbf{x} + \mathbf{b}_\mathbf{x}$, where $\mathbf{b}_\mathbf{x} \in \mathbb{R}^C$ can be thought of as a bias term.

A DN exhibits *generalisation* if it can correctly classify inputs from outside the training set. A DN is *robust* if its predictions are not susceptible to small perturbations in the input.

On the one hand, the generalisation capacity of a DN is a result of the feature learning regime of DN training, with delayed generalisation attributed to a transient phase of linear learning (Lyu et al., 2024; Rubin et al., 2024; Kumar et al., 2024).

On the other hand, the robustness of a DN has been connected to the DN's Jacobians (Rifai et al., 2011; Jakubovitz & Giryes, 2018; Hoffman et al., 2019; Etmann et al., 2019; Chan et al., 2020) and functional geometry (Humayun et al., 2024).

**Functional Geometry.**    Using spline approximation theory (Lyche & Schumaker, 1975; Schumaker, 2007), DNs can be approximated to arbitrary precision or even characterised exactly using splines (Balestriero & Baraniuk, 2018). In particular, there exists a continuous piecewise affine (CPA) approximation to a DN of the form $f(\mathbf{x}) \approx \mathbf{A}_{\omega_{\boldsymbol{\nu}(\mathbf{x})}} \mathbf{x} + \mathbf{b}_{\omega_{\boldsymbol{\nu}(\mathbf{x})}}$, where $\mathbf{A}_{\omega_{\boldsymbol{\nu}(\mathbf{x})}} \in \mathbb{R}^{C \times d}$ and $\mathbf{b}_{\omega_{\boldsymbol{\nu}(\mathbf{x})}} \in \mathbb{R}^C$ are the affine transformation parameters specific to the *linear region* $\omega_{\boldsymbol{\nu}(\mathbf{x})} \subseteq \mathbb{R}^d$ encompassing $\mathbf{x}$.[1] The affine transformations operating on neighbouring linear regions are such that the overall approximation is continuous.

**Definition 1.** *The functional geometry of a DN $f$ is the disjoint of union of $\{\omega_{\boldsymbol{\nu}}\}_{\boldsymbol{\nu} \in \mathcal{V}}$.*

The functional geometry of a DN is a partition of the input space into convex polytopes (Balestriero et al., 2019) and can be parametrised by a *power diagram* subdivision using a collection of *centroid-radius* pairs $\{(\mu_{\boldsymbol{\nu}}, \tau_{\boldsymbol{\nu}})\}_{\boldsymbol{\nu} \in \mathcal{V}} \subseteq \mathbb{R}^d \times \mathbb{R}$, such that

$$\omega_{\boldsymbol{\nu}} = \left\{ \mathbf{x} \in \mathbb{R}^d : \boldsymbol{\nu} = \arg \min_{\boldsymbol{\nu}' \in \mathcal{V}} \left( \|\mathbf{x} - \mu_{\boldsymbol{\nu}'}\|_2^2 - \tau_{\boldsymbol{\nu}'} \right) \right\}. \tag{1}$$

**Theorem 2.** *For a DN $f$, we have $\mu_\mathbf{x} = (\mathbf{J}_\mathbf{x}(f))^\top \mathbf{1}$. (Proof in Appendix H).*

Theorem 2 is significant as it enables us to access the centroids of a DN through an efficient Jacobian-vector product computation. In particular, the centroids for the linear regions can be computed on a per-region basis, despite the functional geometry of a DN being an inherently combinatorial object (Montúfar et al., 2014).

**Neural Tangent Kernel.**    Suppose a DN $f$ has parameters $\theta$. Then the neural tangent kernel (Jacot et al., 2018) between $\mathbf{x}, \mathbf{x}' \in \mathbb{R}^d$ is taken to be $\Theta(\mathbf{x}, \mathbf{x}') = \nabla_\theta f_\theta(\mathbf{x}) (\nabla_\theta f_\theta(\mathbf{x}'))^\top$. The *linear* and *feature* learning regimes of DN training are characterised by having relatively constant or dynamic

---

[1]Here $\boldsymbol{\nu}(\mathbf{x})$ identifies the equivalence class of $\mathbf{x}$ under the collection of all equivalence classes $\mathcal{V}$ constructed by $\sim$ where $\mathbf{x}_1 \sim \mathbf{x}_2$ if and only if $\mathbf{x}_1$ and $\mathbf{x}_2$ are in the same linear region.

neural tangent kernels, respectively (Chizat et al., 2019; Woodworth et al., 2020; Moroshko et al., 2020). The former identifies when the DN approximates a linear function, whereas the latter involves the DN's nonlinearities.

## 2.2 GENERALISATION THROUGH CENTROID ALIGNMENT

Prior work has demonstrated that a simplification of a DN's functional geometry coincides with the onset of grokking and emergent robustness (Humayun et al., 2024). The observed simplification can be categorized as an expansion of the linear regions in which the training data lie, and a concentration of linear regions along the decision boundaries in the input space. This corroborates the general consensus that training DNs with stochastic gradient descent has an implicit bias for margin-maximising solutions (Lyu & Li, 2020; Lyu et al., 2021).

**Definition 3.** *A DN is **centroid-aligned** at $\mathbf{x} \in \mathbb{R}^d$ if $\mu_{\mathbf{x}} = c\mathbf{x}$ for some constant $c \in \mathbb{R}$.*

The geometrical consequences of centroid alignment are visualised vividly in Figure 1, providing initial evidence that considering centroid alignment will be useful for understanding the destination of DN training dynamics.

We can gain some useful insights by exploring the centroid alignment of a toy two-layer, scalar-output DN of the form $f_\theta(\mathbf{x}) = \mathbf{W}^{(2)}\left(\sigma\left(\mathbf{W}^{(1)}\mathbf{x}\right)\right)$, where $\mathbf{W}^{(2)} \in \mathbb{R}^{1 \times d^{(1)}}$, $\mathbf{W}^{(1)} \in \mathbb{R}^{d^{(1)} \times d}$, and $\sigma$ is the ReLU nonlinearity. In particular, we will suppose the DN is being trained using full-batch gradient descent with a learning rate of $\eta$ under the cross-entropy loss function. In Appendix F, we consider vector-output DNs, where we draw analogous conclusions.

**Theorem 4.** *In the setting described above, we have $\partial_t\left(\langle\mathbf{x}, \mu_{\mathbf{x}}\rangle\right) = \frac{\eta}{m}\sum_{p=1}^m \Theta\left(\mathbf{x}, \mathbf{x}_p\right) m_{\mathbf{x}_p}$, where $m_{\mathbf{x}_p} = y_p - \frac{1}{1+\exp(-f_\theta(\mathbf{x}_p))}$. (Proof in Appendix H).*

Theorem 4 says that the inner product between some point $\mathbf{x}$ in the input space and its corresponding centroid $\mu_{\mathbf{x}}$ is a weighted sum of the neural tangent kernel of the point with the points in the training data. Since we would expect $\Theta$ and $m$ to be uncorrelated, a changing inner product implies a dynamic neural tangent kernel, which corresponds to the feature learning regime of training.

Since centroid alignment can only emerge through a changing inner product, we have that centroid-aligned DNs are an artefact of the feature learning regime of training. Conversely, if the norms of the centroids are controlled, then a changing inner product will translate into centroid alignment.

The reason for desiring alignment over just an increasing inner product stems from the geometrical implications. Alignment of the centroids ensures that the linear regions of the input space partition maintain a coherent structure with the overall semantics of the input space (see Figure 10). Maintaining this coherent structure facilitates the attainment of a margin-maximising solution. Therefore, not only is alignment preferred, but it also complements the underlying implicit bias of stochastic gradient descent.

## 2.3 ROBUSTNESS THROUGH JACOBIAN ALIGNMENT

Having established that centroid-aligned DNs exhibit generalisation, we now turn our attention to the Jacobians of the DN, through Theorem 2, to demonstrate their robustness.

**Definition 5.** *A DN is **Jacobian-aligned** at $\mathbf{x} \in \mathbb{R}^d$ if $\mathbf{J}_{\mathbf{x}}(f) = \mathbf{c}\mathbf{x}^\top$ for some vector $\mathbf{c} \in \mathbb{R}^d$.*

Aligned Jacobians are rank-one and are such that each row of the Jacobian is cosine similar to the input point. Etmann et al. (2019); Chan et al. (2020) only consider the alignment of singular rows of a DN's Jacobian (i.e., the row that corresponds to the class of the input). Definition 5 necessitates that all rows of the Jacobian are aligned to the input point, establishing it as a novel property of DNs.

**Corollary 6.** *A Jacobian-aligned DN is centroid-aligned. (Proof in Appendix H).*

Corollary 6 establishes that Jacobian alignment is a stronger property than centroid alignment, meaning, from Section 2.2, that Jacobian-aligned DNs exhibit generalisation. However, with this refined structure comes improved properties.

**Theorem 7.** *For a DN $f$, suppose that $\mathbf{J}_{\mathbf{x}}$ is rank one, of fixed Frobenius norm, and $\mathbf{b}_{\omega_{\mathbf{x}}} = \mathbf{0}$. Then the affine mapping on the linear region $\omega_{\mathbf{x}}$ of the DN's CPA approximation requires the largest*

*perturbation, in terms of $\ell_2$ norm, to misclassify $\mathbf{x}$ when $\mathbf{J_x} = \mathbf{cx}^\top$ for $\mathbf{c}$ having maximum entry at the index of the class of $\mathbf{x}$. (Proof in Appendix H).*

Theorem 7 demonstrates that Jacobian-aligned DNs — in the low-rank regime — are optimally robust. Consequently, we can interpret Jacobian alignment as a refinement of the structure of the DN from a state exhibiting generalisation to a state *also* exhibiting robustness.

A point to note in Theorem 7 is the requirement of the Jacobians to be of fixed Frobenius norm. This prevents increasing the Frobenius norm arbitrarily to achieve robustness. This would result in an increasing inner product between the input and the row of the Jacobian corresponding to the correct class of the input. Whilst a valid solution for a *robust* DN, we can no longer make the connection to centroid alignment to ensure it exhibits generalisation.

Moreover, increasing the norms of the Jacobian arbitrarily will inevitably lead to large centroids, which, due to the nature of the power diagram subdivision Equation (1) and Theorem 15, will lead to overly large linear regions (see Figure 10). This will potentially degrade the expressivity of the DN due to the smaller number of linear regions (Raghu et al., 2017).

Therefore, at least when the Jacobians of the DN are rank one, Jacobian-alignment represents the *optimal* destination where the DN exhibits both generalisation and robustness.

## 3   REACHING THE DESTINATION

In this section, we study how current training regimens attempt to reach the Jacobian-aligned destination of DN training. In Section 3.1, we support the prior theory by demonstrating that, in the regime of DNs with rank one Jacobians, the destination is uniquely characterised by Jacobian-aligned DNs. Then, through empirical simulations, we extend this conclusion to DNs whose Jacobians are not necessarily rank one. However, we then demonstrate in Section 3.2 that standard training protocols, such as weight decay, provide indirect guidance to this destination.

### 3.1   FROM THE LOW-RANK REGIME AND BEYOND

Jacobian-aligned DNs inhabit a class of DNs with rank-one Jacobians at the training data, and Theorem 7 ensures their superior robustness within this class. Therefore, we first corroborate our theory for a two-layer scalar-output ReLU DN classifying XOR cluster data, which trivially has Jacobians of rank one. In this case, the Jacobian is just a vector and thus coincides with the centroid. Throughout training, we track the alignment of a data point in the training set.

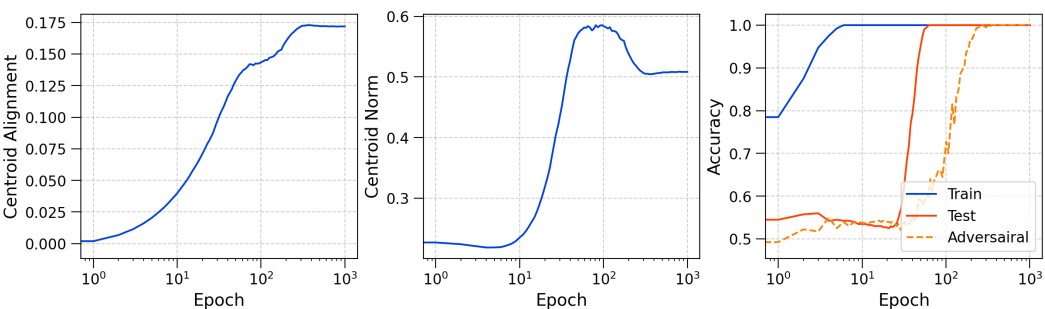

Figure 2: Alignment correlates with the generalisation and onset of robustness for a two-layer scalar-output ReLU DN classifying XOR cluster data. For a point in the training set, we monitor its centroid alignment and centroid norm with the **left** and **centre** plot, respectively. In the **right** plot, we monitor the accuracy and robustness of the DN. For further experimental details refer to Appendix A.1.

In Figure 2, we observe that as the DN memorises, alignment does not increase; whereas, during generalisation, alignment increases as expected by Theorem 4. After a slight plateau in alignment, a further increase in alignment correlates with the onset of robustness. Therefore, it appears that Jacobian alignment is the destination of these dynamics.

To go beyond DNs with rank one Jacobians, we can leverage prior work that identifies an implicit bias of stochastic gradient descent to regularise for low rank weight matrices (Le & Jegelka, 2022; Huh et al., 2023; Timor et al., 2023; Yunis et al., 2024b; Galanti et al., 2025). We empirically observe that this holds for the Jacobian matrices too (see Figure 3), and thus we would expect that alignment becomes an increasingly necessary condition for robustness. Indeed, in Figure 4 (left), we see that the alignment of a DN training on the MNIST classification task increases during grokking.

Thus, we conclude that in the more general setting where Jacobians are not necessarily rank one, Jacobian alignment is what the DNs want to *learn*. We support this with Appendix B, where we show that if we prevent the DNs from attaining Jacobian-alignment it fails to learn.

### 3.2 STANDARD TRAINING PROTOCOLS PROCEED INDIRECTLY TOWARD ALIGNMENT

Our hypothesis here is that standard training protocols fail to control the norms of the Jacobian of a DN, and this delays the onset of alignment. In particular, there is too much initial optimisation pressure to achieve generalisation and robustness through increasing inner products, rather than optimising for alignment. Indeed, in Figure 2, we see an initial increase in the norms of the centroids that eventually curtails the alignment of the DN and prevents the onset of robustness. However, the norms of the centroid eventually wither, allowing the DN to become robust, highlighting that standard training protocols still guide the DN to alignment. Although this provides empirical evidence that it is important to control the norms of the Jacobians to induce alignment, we can theoretically demonstrate this with the following result.

**Theorem 8.** *Let $\ell$ be a convex, non-negative, and differentiable function. Then minimising $\mathcal{L}$ by optimising over $\mathbf{J}_{\mathbf{x}_p}$ under the constraint that $\left\|\mathbf{J}_{\mathbf{x}_p}\right\|_F^2 \leq \alpha$ for $p = 1, \ldots, m$, yields a DN that is Jacobian-aligned on the training set. (Proof in Appendix H).*

Theorem 8 demonstrates that Jacobian-aligned DNs minimise the training objective under the constraint that the Frobenius norms of the Jacobian matrices of the DN are bounded. In particular, Theorem 8 considers a convex optimisation problem over the Jacobians. In practice, DNs are optimised over their weights in a non-convex setting, where multiple weight configurations may correspond to Jacobian-aligned states. Nevertheless, Theorem 8 tells us that, by controlling the norms of the Jacobians of the DN, there is optimisation pressure to induce alignment since it will optimise the training objective.

Yang et al. (2020) has also observed that standard training procedures do not effectively control the norms of the Jacobian of the DN. This explains the current confusion surrounding the significance of weight-decay in the grokking phenomenon (Power et al., 2022; Nanda et al., 2022; Varma et al., 2023; Lee et al., 2024; Kumar et al., 2024), as weight-decay only weakly controls the norms of the Jacobians.

All of this leads us to the claim that *Jacobian alignment explains grokking* (i.e., delayed generalisation), in the sense that *Jacobian alignment corresponds to the destination of DN training dynamics and standard training pipelines only weakly induce it.* Next, we empirically support this hypothesis by introducing a Jacobian regularisation strategy and observing its effects on alignment and grokking.

## 4 GROKALIGN: ACCELERATING AND IMPROVING ALIGNMENT

We have established that DNs want to be Jacobian-aligned and that standard training strategies are ineffective at inducing alignment. To remedy this, we introduce the *GrokAlign* regularisation strategy. GrokAlign significantly accelerates the grokking of DNs and improves the robustness of DNs by guiding them to the Jacobian-aligned destination.

### 4.1 GROKALIGN ALGORITHM

GrokAlign explicitly enforces the Jacobian norm constraint of Theorem 8 by appending the average Frobenius norm of the Jacobian matrices of the training data[2] to the loss function with some weighting coefficient, $\lambda_{\text{Jac}}$. That is, $\mathcal{L}_{\text{GrokAlign}} = \mathcal{L}_{\text{Class}} + \lambda_{\text{Jac}} \frac{1}{m} \sum_{p=1}^{m} \left\|\mathbf{J}_{\mathbf{x}_p}\right\|_F^2$. Consequently, with sufficient

---

[2]In practice, we utilise some approximation of this derived in Hoffman et al. (2019).

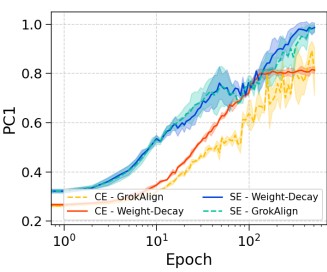

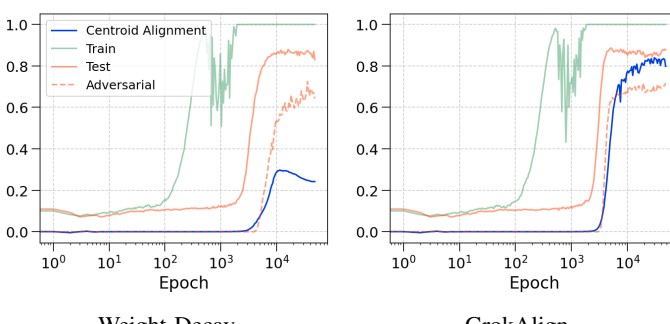

Weight-Decay                    GrokAlign

Figure 3: In multiple training regimes, the effective rank of the Jacobian matrices evaluated at the training data tends towards rank one (evidenced by higher PC1 values). For more experimental details, see Appendix A.2.

Figure 4: A fully connected DN training on the MNIST classification task using GrokAlign achieves 60% adversarial accuracy in 62% fewer epochs. Moreover, GrokAlign leads to a significantly greater alignment in the DN. For more experimental details, see Appendix A.3.

pressure to constrain the norm of the Jacobians, provided we continue optimising the training objective, Theorem 8 says that we should achieve alignment.

Reassuringly, similar forms of Jacobian regularisation have been demonstrated to improve the robustness of DNs in prior work (Rifai et al., 2011; Gu & Rigazio, 2014; Jakubovitz & Giryes, 2018; Hoffman et al., 2019).

Figure 4 (right) provides initial evidence that GrokAlign successfully induces alignment and accelerates grokking. In particular, in comparison to the DN trained with weight decay, the DN trained with GrokAlign achieves a greater alignment, generalisation and robustness. Since weight decay ineffectively controls the norms of the DN's Jacobians, this vividly emphasises the points made at the end of Section 2; namely, that although increasing the inner product between centroids and their input point presents a valid destination, it is not optimal.

## 4.2 EXPERIMENTS: ACCELERATING GROKKING

We compare the effectiveness of GrokAlign to two other methods designed to accelerate grokking against a baseline. Grokfast (Lee et al., 2024) works to accelerate the rate of grokking by manipulating the gradients during training to amplify certain signals. OrthoGrad (Prieto et al., 2025) aligns gradients to prevent naïve loss minimisation and encourage generalisation.[3]

We apply these methods to the XOR grokking setup of Section 3.1, a fully connected DN training on MNIST (Liu et al., 2022), a fully connected DN performing modular addition (Mallinar et al., 2025), and a fully connected DN training on a sparse parity task (Prieto et al., 2025). Across multiple random initialisations, we measure the number of epochs required to reach the grokked state, as specified in Table 3. To determine statistical significance against the baseline, we perform a paired t-test on the number of epochs required to grok.

From Table 1, we see that GrokAlign provides the most significant acceleration of grokking, only once being less effective than OrthoGrad and always more effective than Grokfast. In particular, it works consistently across each setting, whereas the other regularisation strategies observe variable performance. We also analyse the reduction in optimisation time in Appendix C, where we draw similar conclusions.

Another proposed strategy for inducing the grokked state involves adversarial training (Tan & Huang, 2024). Since adversarial training covers a broad range of regularisation strategies, we do not consider it in this analysis. In Appendix D.1, we provide an analysis of its connection to GrokAlign.

Furthermore, in Appendix E we explore the application of GrokAlign to one-layer transformers trained on modular addition (Power et al., 2022). Here we see, under certain conditions, a similar acceleration of grokking.

---

[3]In each method, we do not change the weight decay of the training procedure.

Table 1: GrokAlign consistently accelerates the rate of grokking most effectively compared to Grokfast and OrthoGrad, two well-known regularisation strategies for accelerating grokking. For each setting, we consider the training pipelines across twenty-five random initialisations and record the epoch at which the grokked state is reached. We report the mean value of epochs, the rate of improvement over the baseline training pipeline, and the $p$-value of the corresponding paired t-test performed between the individual training runs of the considered regularisation and the baseline. For the setup involving a DN training on MNIST, we consider training the DN with both the cross-entropy loss function and the squared error loss function. On the modular addition task and the MNIST with squared error, we found that OrthoGrad leads to unstable training, and thus we excluded them from the analysis. For more experimental details, see Appendix A.4.

| XOR | Baseline | Grokfast | OrthoGrad | GrokAlign |
|---|---|---|---|---|
| Number of Epochs | 154 | 154 | **37** | 101 |
| Rate of Speed-Up | – | 1.0 | **4.12** | 1.52 |
| $p$-value | – | – | $1 \times 10^{-30}$ | $2.5 \times 10^{-17}$ |

| Sparse Parity | Baseline | Grokfast | OrthoGrad | GrokAlign |
|---|---|---|---|---|
| Number of Epochs | 2128 | 1888 | 256 | **110** |
| Rate of Speed-Up | – | 1.13 | 8.33 | **19.32** |
| $p$-value | – | 0.063 | $8.6 \times 10^{-13}$ | $1 \times 10^{-13}$ |

| MNIST - Cross Entropy | Baseline | Grokfast | OrthoGrad | GrokAlign |
|---|---|---|---|---|
| Number of Epochs | 2508 | 2364 | 2676 | **328** |
| Rate of Speed-Up | – | 1.06 | 0.94 | **7.65** |
| $p$-value | – | 0.10 | – | $6.7 \times 10^{-16}$ |

| MNIST - Squared Error | Baseline | Grokfast | OrthoGrad | GrokAlign |
|---|---|---|---|---|
| Number of Epochs | 7496 | 7200 | – | **1154** |
| Rate of Speed-Up | – | 1.04 | – | **6.50** |
| $p$-value | – | 0.02 | – | $4.02 \times 10^{-22}$ |

| Modular Addition | Baseline | Grokfast | OrthoGrad | GrokAlign |
|---|---|---|---|---|
| Number of Epochs | 268 | 254 | – | **200** |
| Rate of Speed-Up | – | 1.06 | – | **1.34** |
| $p$-value | – | $3.5 \times 10^{-12}$ | – | $4.8 \times 10^{-19}$ |

## 4.3 EXPERIMENTS: IMPROVING ROBUSTNESS

In this section, we consider the delayed robustness phenomenon presented in Humayun et al. (2024) on a convolutional neural network learning the CIFAR10 (Krizhevsky & Hinton, 2009) image classification task. Throughout training, we monitor the alignment (i.e. cosine similarity) and inner product of input points to the row of the Jacobian corresponding to the class of the input points or their corresponding centroids.

The main observations from Figure 5 are the following: (1) An initial increase in centroid alignment correlates with the onset of generalisation as exhibited by the increasing test accuracy. (2) Both DNs progress toward solutions where the inner product values increase; however, the DN trained with GrokAlign does this more effectively through alignment. The vertical grey dashed line makes it evident that delayed robustness emerges in the GrokAlign DN when the Jacobian alignment of the model increases. Note that we can still lose robustness even when the Jacobian alignment is increasing, since the DN may not be in a low rank regime. (3) The final robustness of the DN trained with GrokAlign is higher than that trained with weight decay. We support this by noting that the DN trained with GrokAlign has an accuracy of 63.6% on CIFAR10-C (Hendrycks & Dietterich, 2019), whereas the one trained with weight decay has an accuracy of 61.8% on CIFAR10-C. We break down the nature of this robustness in Table 5. This is expected from the discussions of Section 2, where *alignment* is identified as preferable over just increasing inner products. Indeed, toward the end of training, the DN trained with weight decay sees a degrading performance. (4) Centroid alignment emerges as an efficient indicator – computing centroids for Figure 5 added less than two wall-clock minutes to each training time, which were on the order of hours – of progress toward the *optimal* destination. Indeed, we can correlate the rise in centroid alignment of the DN trained with GrokAlign with the performance improvements of the DN. Based on the trajectory of centroid alignment, we can see that we could continue to train this DN and continue to experience an improvement in its generalisation and robustness without worrying about overfitting.

## 5 DISCUSSION

In this paper, we have made progress towards understanding the destination of DN training dynamics beyond extrinsic metrics like training error. We leveraged prior work to note that this destination

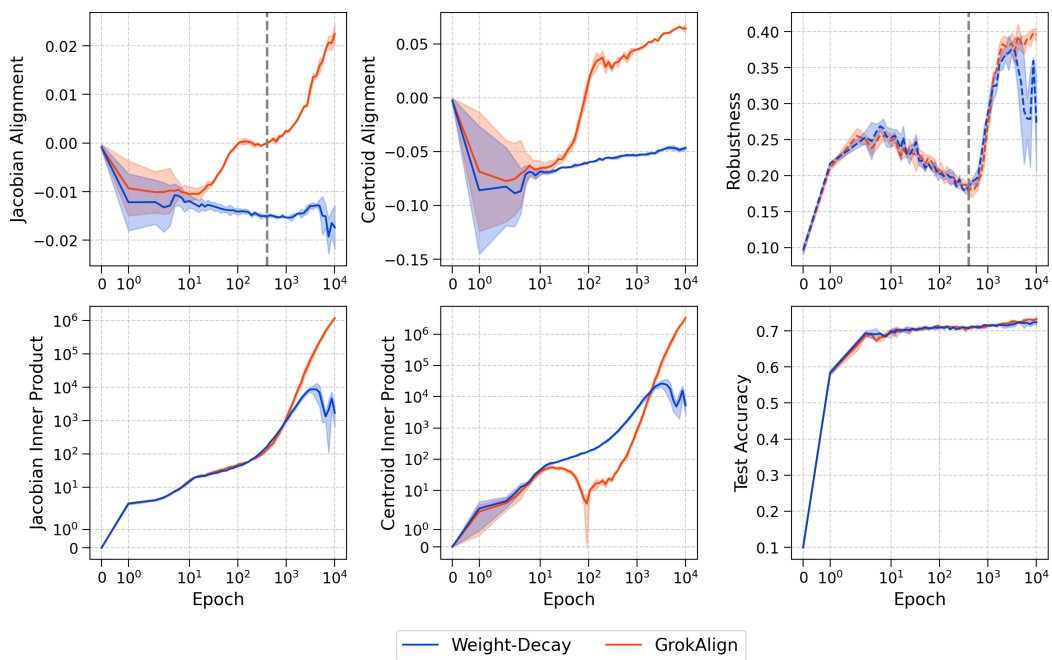

Figure 5: Here, we study the alignment of a convolutional neural network trained on the CIFAR10 classification task. Jacobian alignment identifies the onset of delayed robustness, with GrokAlign enhancing the saliency of this alignment to lead the DN to a destination of improved robustness. GrokAlign achieves a maximum robustness of $39.8\%$ compared to $37.6\%$ for weight decay. Jacobian alignment (resp. inner product) refers to the cosine similarity (resp. inner product) between the training samples and the row of the Jacobian corresponding to the class of the sample. Centroid alignment (resp. inner product) refers to the cosine similarity (resp. inner product) between the training samples and their corresponding centroid. Robustness is measured by perturbing samples from the test set with ten steps of PGD (Madry et al., 2018) with $\alpha = \frac{2}{255}$ and $\epsilon = \frac{4}{255}$. We present the average statistics across three random initialisations for each training protocol. For further experimental details, see Appendix A.5.

can be identified as a simplification of the DN's geometry. In particular, this simplification is an expansion of the linear regions of the continuous piecewise approximation to the DN. This supports a prior consensus in the literature that gradient descent has an implicit bias for margin-maximising solutions.

To formalise this intuition, we utilised the centroids of a DN, which are known to parametrise these linear regions. We showed that an alignment (in terms of cosine similarity) of these centroids to the input points in the corresponding linear regions describes the onset maximal generalisation in DNs. Refining the centroid alignment property to Jacobian alignment then yields robust DNs. By supporting these theoretical results with empirical observations, we conclude that Jacobian alignment is the destination of DN training dynamics.

Although standard training pipelines seek Jacobian alignment, they are indirect. Therefore, we introduce GrokAlign as a regularisation strategy that is guaranteed to induce Jacobian alignment through optimising the training objective. Indeed, we observe that GrokAlign significantly accelerates the rate of grokking and improves the robustness of DNs.

One limitation of our investigation is the dependency on the assumption that gradient descent has an implicit bias to minimise the rank of the weight matrices of the DN. A deeper characterisation of this phenomenon and its relationship to the Jacobians of the DN would help support our conclusions.

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

# A   EXPERIMENTAL DETAILS

In Table 2, we elaborate on the experimental details and provide the computation time necessary to perform the experiments.

Table 2: Here we detail the hardware that was used to perform our experiments, along with the time taken to run the experiments.

| Experiment | Hardware | Computation Time (hours) |
|---|---|---|
| Figure 2 | TITAN X | 1 |
| Figure 3 | TITAN X | 10 |
| Figure 4 | GTX 1080Ti | 6 |
| Tables 1 and 4 | GTX 1080Ti | 50 |
| Figure 5 | RTX 8000 | 50 |
| Figure 6 | TITAN X | 83 |
| Figure 7 | TITAN X | 125 |
| Figure 8 | RTX 8000 | 12 |

## A.1   FIGURE 2

The setup is similar to that of Xu et al. (2025), entailing a scalar-output two-layer fully connected network learning on XOR cluster data. The XOR cluster data contains 40000-dimensional vectors of the form $\mathbf{x} = \left(x_1, x_2, \tilde{\mathbf{x}}^\top\right)^\top \in \mathbb{R}^{40000}$, where $x_1, x_2 \in \{\pm 1\}$ and $\tilde{\mathbf{x}} \in \mathbb{R}^{39998}$. The 400 samples used to train the network are constructed by sampling entries $x_1$, $x_2$ uniformly from $\{\pm 1\}$ and entries of $\tilde{\mathbf{x}}$ uniformly from $\{\pm \epsilon\}$, here we take $\epsilon = 0.05$. The corresponding label of such a sample is $x_1 x_2 \in \{\pm 1\}$. A similar sample of the same size is generated as a test set.

The DN is trained over 1000 epochs using full-batch gradient descent with a learning rate of 0.1 and a weight-decay of 0.1. To test the adversarial accuracy of the DN, we perturb the last 39998 components of the test set with random noise of standard deviation 0.2.

## A.2   FIGURE 3

Here we train DNs with two hidden layers of widths 256 on the MNIST classification task (Lecun et al., 1998). Either under the squared-error loss function or the cross-entropy loss function using the AdamW optimiser (Loshchilov & Hutter, 2019) with a learning rate of 0.001 and a batch size of 196. Weight decay and GrokAlign are both applied with constants 0.001. Throughout training, we recorded the average explained variance of the first principal component of the Jacobians evaluated at the training data (PC1), namely $\frac{\sigma_1^2}{\sum_{i=1}^{r} \sigma_i^2}$ where $\sigma$ are the singular values of the Jacobian. When this normalised value equals one, the Jacobian matrix is rank one. We repeat these training runs across three random initialisations and report the mean and standard deviations.

## A.3   FIGURE 4

Here we adopt a setup similar to that of Liu et al. (2022). That is, we train a three-hidden-layer DN on a random 1024-sample subset of the MNIST classification task. The DN has a constant width of 196, no bias terms and the weights are multiplied by a factor of eight on initialisation. The DN is trained with the AdamW optimiser at a learning rate of 0.001, a batch-size of 128 and a weight-decay of 0.01. When applied, GrokAlign is used with $\lambda_{\text{Jac}}$ equal to 0.001.

Centroid alignment is computed as an average of the samples from the training set, and adversarial accuracy is evaluated by computing the accuracy of the model when ten steps of PGD (Madry et al., 2018) with a step size of $\frac{4}{255}$ are applied to the test set.

## A.4  TABLE 1

This experiment consists of different training setups, and we will detail each one in turn. In Table 3 we provide the grokked state criteria used for each setup. Throughout each configuration, we use GrokFast-EMA with $\alpha = 0.8$ and $\lambda = 0.1$.

Table 3: Here we state the criteria used to identify the grokked state of a DN for the experiments of Tables 1 and 4.

| Setup | Criterion |
|---|---|
| XOR | Test accuracy and adversarial accuracy greater than $95\%$ |
| Sparse Parity | Test accuracy greater than $90\%$ |
| MNIST - Cross Entropy | Test accuracy greater than $80\%$ |
| MNIST - Squared Error | Test accuracy greater than $80\%$ |
| Modular Addition | Test accuracy greater than $99\%$ |

**XOR.**  Refer to Appendix A.1.

**Sparse Parity.**  This setup is taken from Prieto et al. (2025). It involves performing the binary classification of a bit string based on the parity of the sum of a select few indices. More specifically, the training distribution constitutes 2000 samples of bit strings of length 40 whose labels are given by the parity of the sum of the first three bits. We train a DN with two hidden layers of widths 200 on half this training distribution, and use the other half to test the DN. The DN is trained under the cross-entropy loss function using the AdamW optimiser with a learning rate of 0.01. Weight-decay is applied at 0.1 and GrokAlign is used with $\lambda_{\text{Jac}}$ equal to 0.1.

**MNIST.**  This constitutes the same setup as detailed in Appendix A.3, except we consider additionally using the cross-entropy loss function. Moreover, when applying GrokAlign, we do so with $\lambda_{\text{Jac}}$ equal to 0.01.

**Modular Addition.**  This setup is similar to one from Mallinar et al. (2025) and involves a one hidden layer fully connected DN learning addition modular 61. The DN has a width 256 and uses a quadratic activation function. The DN is trained using the AdamW optimiser with a learning rate of 0.001 and a batch size of 32. A weight decay of 1.0 and GrokAlign is used with $\lambda_{\text{Jac}}$ equal to 0.01.

## A.5  FIGURE 5

Here we train a convolutional neural network with five convolutional layer feature extractors and a two-layer fully connected classifier. The convolutional layers have filters starting at 32 and then expanding by a factor of two for successive layers. The fully connected classifier has a width 256. Each layer uses the ReLU activation function.

The DN is then trained using the AdamW optimiser with a learning rate of 0.001 and a batch size of 200. Weight decay is applied at 0.001 and GrokAlign is applied with $\lambda_{\text{Jac}}$ equal to 0.001.

## A.6  FIGURE 8

The training pipeline is identical to that of Nanda et al. (2022). Except we utilise GrokAlign with $\lambda_{\text{Jac}}$ equal to 0.001.

## B  GROK(UN)ALIGN

To further support the influence GrokAlign has on grokking, we can consider its effect on a DN when applied in reverse, Grok(Un)Align. That is, we can set $\lambda_{\text{Jac}}$ to a negative quantity to explicitly exit the constrained optimisation regime of Theorem 8. In Figure 6, we see that Grok(Un)Align prevents the norms of the centroids from decreasing. Thus, during generalisation, we do not get any alignment.

Subsequently, the DN collapses since in this low-Jacobian-rank regime, it can only be robust through alignment. This demonstrates how the Jacobian-aligned state is the only destination available to DNs that, more generally, do not necessarily have low rank Jacobians a priori.

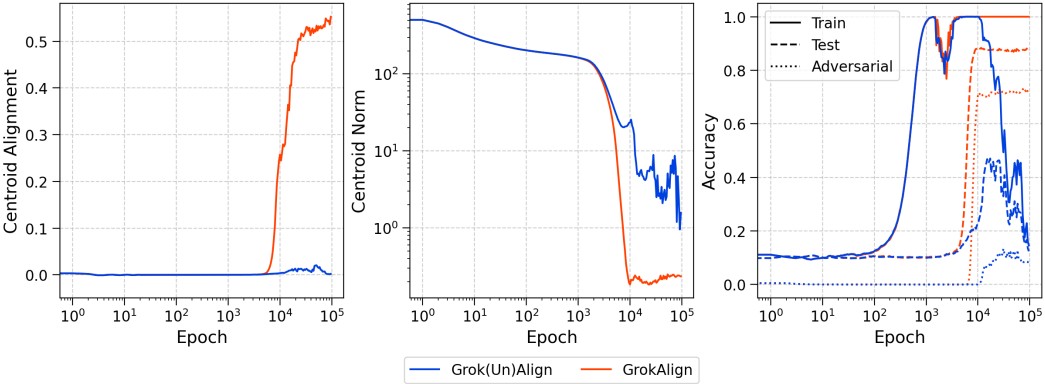

Figure 6: Here we adopt the same setup as Figure 4. We either regularise using GrokAlign with $\lambda_{\text{Jac}}$ equal to $0.001$ or using GrokAlign with $\lambda_{\text{Jac}}$ equal to $-0.001$ – referred to as Grok(Un)Align. In addition to centroid alignment and accuracy, we record the Euclidean norm of the centroids in the **centre** plot. We repeat these training runs across four random initialisations and report the mean trajectories.

## C  ACCELERATING GROKKING TIME

Since GrokAlign references the DN's Jacobians during training, it inevitably incurs additional computational burden over other regularisation techniques. Therefore, although we demonstrated with Table 1 that GrokAlign reduces the number of epochs required to reach the grokked state, it may be the case that wall-clock time is not substantially improved with GrokAlign. Fortunately, with Table 4, we demonstrate that the acceleration of the optimisation epochs largely translates to an acceleration in wall-clock optimisation time.

Table 4: The experimental details and presentation of results match that of Table 1, except here we present the wall-clock times of the training trajectories as measured in seconds.

| XOR Grokking | Baseline | Grokfast | OrthoGrad | GrokAlign |
|---|---|---|---|---|
| Time | 43.95 | 44.73 | **10.99** | 30.09 |
| Rate of Speed-Up | – | 0.98 | **4.00** | 1.46 |
| $p$-value | – | – | $8.1 \times 10^{-27}$ | $4.9 \times 10^{-15}$ |

| Sparse Parity | Baseline | Grokfast | OrthoGrad | GrokAlign |
|---|---|---|---|---|
| Time | 4.13 | 4.75 | 0.86 | **0.45** |
| Rate of Speed-Up | – | 0.87 | 4.82 | **9.15** |
| $p$-value | – | – | $5.1 \times 10^{-12}$ | $2.5 \times 10^{-13}$ |

| MNIST - Cross Entropy | Baseline | Grokfast | OrthoGrad | GrokAlign |
|---|---|---|---|---|
| Time | 2965.81 | 2801.78 | 3112.29 | **467.24** |
| Rate of Speed-Up | – | 1.06 | 0.95 | **6.35** |
| $p$-value | – | 0.10 | – | $9.4 \times 10^{-16}$ |

| MNIST - Squared Error | Baseline | Grokfast | OrthoGrad | GrokAlign |
|---|---|---|---|---|
| Time | 7932.45 | 7614.97 | – | **1478.13** |
| Rate of Speed-Up | – | 1.04 | – | **5.37** |
| $p$-value | – | 0.03 | – | $9.84 \times 10^{-21}$ |

| Fully-Connected Modular Addition | Baseline | Grokfast | OrthoGrad | GrokAlign |
|---|---|---|---|---|
| Time | 350.41 | **315.48** | – | 414.16 |
| Rate of Speed-Up | – | **1.11** | – | 0.85 |
| $p$-value | – | $1.5 \times 10^{-8}$ | – | – |

# D    THE ROBUSTNESS OF GROKALIGN

## D.1    GROKALIGN AND ADVERSARIAL TRAINING

In Tan & Huang (2024) demonstrated relationships between the robustness and generalisation of a DN. Leading to the idea that adversarial training can accelerate the grokking of a DN. GrokAlign can be seen as a form of adversarial training, where the robustness is induced due to the increased alignment of the subsequent DN.

In Figure 7, we support the idea that the grokked state of a DN, in this setting at least, corresponds to the Jacobian-aligned state. Indeed, adversarial training accelerates the onset of the grokked state whilst also improving the alignment of the DN. Although the extent of the alignment is not the same as that of the DN trained using GrokAlign and the acceleration is not as high. Highlighting how optimising for alignment is the most direct mechanism through which to induce the grokked state of the DN.

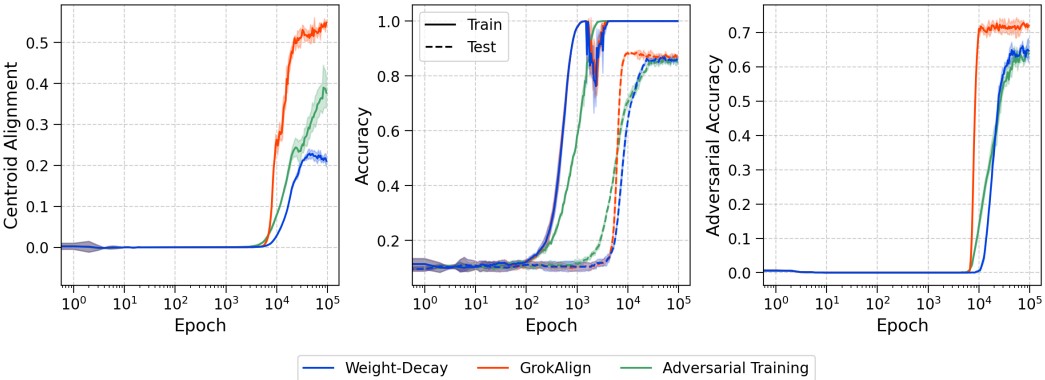

Figure 7: We compare adversarial training to weight-decay and GrokAlign at inducing the robustness and alignment of DNs trained on the MNIST classification task using the squared-error loss function (the same setup as Table 1). We repeat the training for each form of regularisation across three random initialisations and report the mean and standard deviation statistic values.

## D.2    BEYOND ADVERSARIAL PERTURBATIONS

Although Theorem 8 is a statement about robustness with respect to small, noisy perturbations, robustness can also be considered with respect to realistic corruptions, such as contrast or brightness changes to an image. The CIFAR10-C dataset (Hendrycks & Dietterich, 2019) was constructed to test a DN against these sorts of corruptions. In Table 5, we highlight how the DNs of Figure 5 that are Jacobian-aligned, in conjunction with being robust to adversarial perturbations, are more robust to these sorts of corruptions.

# E    ALGORITHMIC GROKKING

One-layer transformer DNs trained on modular addition have also demonstrated grokking (Power et al., 2022). Interestingly, it was determined that the grokked state of this DN involves the execution of an algorithm (Nanda et al., 2022). Thus, this would be an instance in which the grokked state of the DN may not be Jacobian-aligned, since Jacobian-alignment pertains to DN classifiers.

GrokAlign does not accelerate the rate of grokking in this instance (see Figure 8a) since GrokAlign implicitly regularises for a classification-style solution. We can see this by the low Gini coefficients observed under GrokAlign – high Gini coefficients of the embedding matrices are an artefact of the algorithmic-style solution (Nanda et al., 2022). Therefore, if we instead freeze the embedding matrices during training, to inhibit the implementation of the algorithmic-style solution, we observe that GrokAlign accelerates grokking compared to weight-decay (see Figure 8b).

Table 5: Here we break down the performance of the DN's from Figure 5 on the CIFAR10-C dataset.

| Perturbation | GrokAlign | Weight-Decay |
|---|---|---|
| Fog | 59.8% | 56.6% |
| JPEG Compression | 70.5% | 69.4% |
| Zoom Blur | 61.6% | 58.2% |
| Speckle Noise | 65.2% | 64.5% |
| Spatter | 64.7% | 64.1% |
| Shot Noise | 65.3% | 64.5% |
| Defocus Blur | 66.0% | 63.6% |
| Elastic Transform | 63.7% | 61.5% |
| Gaussian Blur | 62.8% | 59.8% |
| Frost | 63.8% | 62.1% |
| Saturate | 68.5% | 68.6% |
| Brightness | 71.0% | 69.5% |
| Snow | 63.3% | 62.4% |
| Gaussian Noise | 62.5% | 61.5% |
| Motion Blur | 60.3% | 57.1% |
| Contrast | 48.5% | 44.8% |
| Impulse Noise | 57.1% | 56.4% |
| Pixelate | 69.8% | 68.6% |

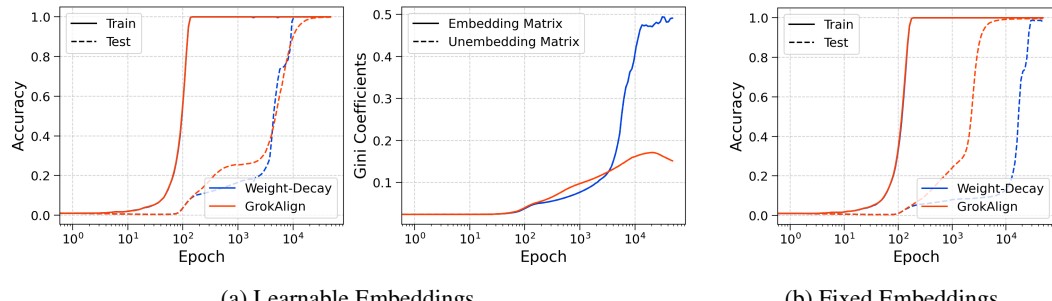

(a) Learnable Embeddings

(b) Fixed Embeddings

Figure 8: Here we train one-layer transformer DNs to perform modular addition. We consider training the transformer with and without GrokAlign. In Figure 8a, we allow the parameters of the embedding layer to be learnable, whereas in Figure 8b we keep the parameters of the embedding layer fixed throughout training. Therefore, we only report Gini coefficient values for the DN with learnable embeddings, as for the DN in Figure 8b these values remain constant. For more experimental details refer to Appendix A.6.

## F    CENTROID DYNAMICS OF VECTOR-OUTPUT DEEP NETWORKS

Consider the case of a general vector output, namely $d^{(2)} \geq 2$, with $\mathcal{L}_{\text{Class}}$ being the cross-entropy loss function or the squared-error loss function. Namely, we consider

$$\ell\left(f\left(\mathbf{x}_p\right), y_p\right) = -\log\left(\frac{\exp\left([f\left(\mathbf{x}_p\right)]_{y_p}\right)}{\sum_{c=1}^{d^{(2)}} \exp\left([f\left(\mathbf{x}_p\right)]_c\right)}\right)$$

for the cross-entropy loss function, or

$$\ell\left(f\left(\mathbf{x}_p\right), y_p\right) = \left\|\mathbf{e}_{y_p} - f\left(\mathbf{x}_p\right)\right\|_2^2$$

for the squared-error loss function.

**Proposition 9.** *In the setting described above, we have*

$$\partial_t \left( \langle \mathbf{x}, \mu_{\mathbf{x}} \rangle \right) = \frac{\eta}{m} \sum_{p=1}^{m} \left( \left( \mathbf{m}_{\mathbf{x}_p}^{\top} W^{(2)} \mathbf{Q}_{\mathbf{x}_p} \mathbf{Q}_{\mathbf{x}} \left( \mathbf{W}^{(2)} \right)^{\top} \mathbf{1} \right) \langle \mathbf{x}, \mathbf{x}_p \rangle \right.$$

$$\left. + \mathbf{x}^{\top} \left( \mathbf{W}^{(1)} \right)^{\top} \mathbf{Q}_{\mathbf{x}} \sigma \left( \mathbf{W}^{(1)} \mathbf{x}_p \right) \mathbf{m}_{\mathbf{x}_p}^{\top} \mathbf{1} \right)$$

*where*

$$\mathbf{m}_{\mathbf{x}_p} = \mathbf{e}_y - \frac{\exp \left( [f_\theta \left( \mathbf{x}_p \right)]_{y_p} \right)}{\sum_{c=1}^{C} \exp \left( [f_\theta \left( \mathbf{x}_p \right)]_c \right)}$$

*in the case of the cross-entropy loss function and*

$$\mathbf{m}_{\mathbf{x}_p} = 2 \left( \mathbf{e}_y - f_\theta \left( \mathbf{x}_p \right) \right).$$

**Corollary 10.** *In the setting of Proposition 9, under the cross entropy loss function, we have*

$$\partial_t \left( \langle \mathbf{x}, \mu_{\mathbf{x}} \rangle \right) = \frac{\eta}{m} \sum_{p=1}^{m} \left( \mathbf{m}_{\mathbf{x}_p}^{\top} \mathbf{W}^{(2)} \mathbf{Q}_{\mathbf{x}_p} \mathbf{Q}_{\mathbf{x}} \left( \mathbf{W}^{(2)} \right)^{\top} \mathbf{1} \right) \langle \mathbf{x}, \mathbf{x}_p \rangle$$

$$:= \frac{\eta}{m} \sum_{p=1}^{m} \frac{\iota_{\mathbf{x},p}}{\|\mathbf{x}_p\|_2} \langle \mathbf{x}, \mathbf{x}_p \rangle$$

That is, in the context of the cross-entropy loss function, the centroids are aligned in a manner that is proportional to the alignment of their encompassing points with the training data. More specifically, with the intuition that the role of $\mathbf{W}^{(2)}$ in $f_\theta$ is to be a collection of filters facilitating the classification of each class, the quantity $\mathbf{W}^{(2)} \mathbf{Q}_{\mathbf{x}_p} \left( \mathbf{W}^{(2)} \mathbf{Q}_\nu \right)^{\top} \mathbf{1}$ can be thought of as identifying how each feature of $\mathbf{x}_p$ correlates with the features of the region $\omega_\nu$. Since $\mathbf{m}_{\mathbf{x}_p}$ is positive on the correct class and negative for the incorrect classes, the term $\iota_{\mathbf{x}',p}$ is largest when $\omega_{\mathbf{x}'}$ has identified features that correlate with the features of $\mathbf{x}_p$ that indicate the class it belongs too. In such a case, the centroid $\mu_{\mathbf{x}'}$ moves in the direction of $\mathbf{x}_p$ to further maximise this correlation. Showing how regions $\omega_{\mathbf{x}'}$ are being optimised to capture the features of classes that help it distinguish itself from the other classes. Therefore, we can see DN training more as a process of allocating linear regions to different features that best distinguish themselves from the other classes. This would suggest that when we observe the centroids of a layer of a DN aligning with the data it encompasses, the DN is performing feature extraction. In particular, the centroid of a training point is incentivised to positively align with itself.

## G  THE GEOMETRY OF CENTROID ALIGNMENT

Centroids have a known connection to the functional geometry of DNs through the spline theory of deep learning (Balestriero & Baraniuk, 2018). They form part of the parameter of a region in the power diagram subdivision formulation of DN's functional geometry (Balestriero et al., 2019). In Figure 1, we demonstrated the implications of centroid alignment on this geometry. More specifically, we showed that it corresponds to the linear regions flattening around inputs of the same class, in a similar way to the region migration phenomenon identified in Humayun et al. (2024). Here we support that claim by illustrating it beyond the input points considered in Figure 1.

Throughout the main text, we emphasised the preference for centroid alignment over merely increasing the centroid inner product. Our reasons were both theoretical and geometrical. On the geometrical side, we stated that a centroid-aligned DN would have a functional geometry that is more semantically coherent to the structure of the input space, which would ensure that a margin-maximising solution would be obtained. Furthermore, it would ensure that none of the linear regions grew too large to affect the expressivity of the DN. Here, we support that by comparing the functional geometries of such DNs.

In Figure 10, we visualise the centroids of Figure 4, where the DN trained using weight-decay yields centroids with large inner products whilst the DN trained using GrokAlign yields aligned centroids. Note how for the DN with large inner products: The centroids have no resemblance to the input

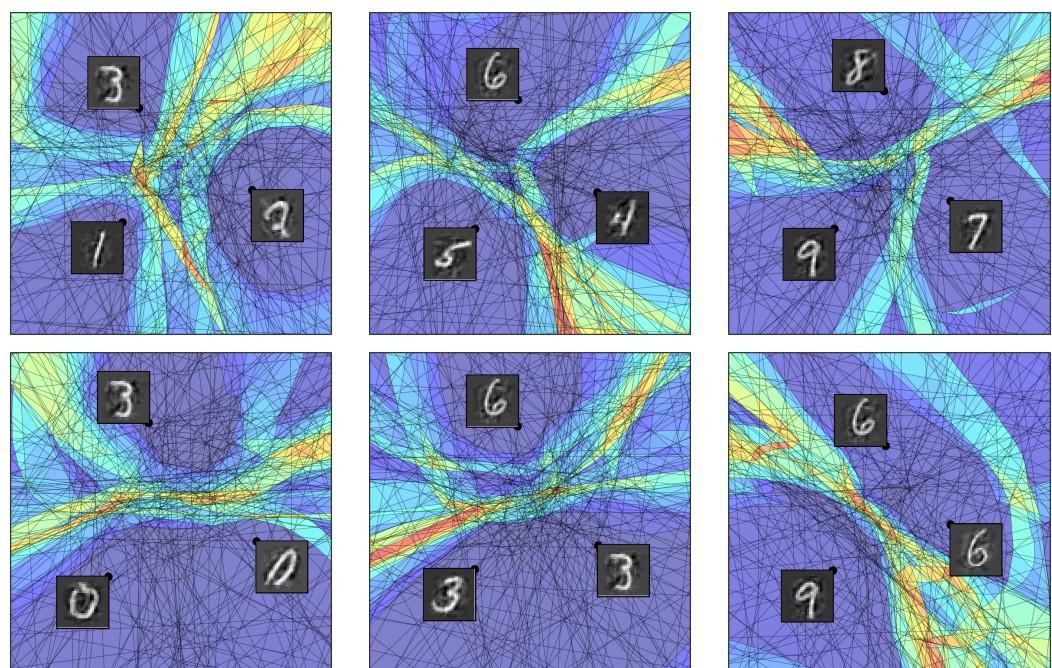

Figure 9: Here we corroborate the right plot of Figure 1 by replicating it on other samples in the input space.

samples. The decision boundaries – which can be matched with the regions with brighter colours – are not evenly placed between the samples. The regions are relatively larger. Each of these indicates why seeking alignment is preferable, since under alignment: The centroids capture the features of the input samples. The DN learns a margin-maximising solution. The DN maintains a higher level of expressivity.

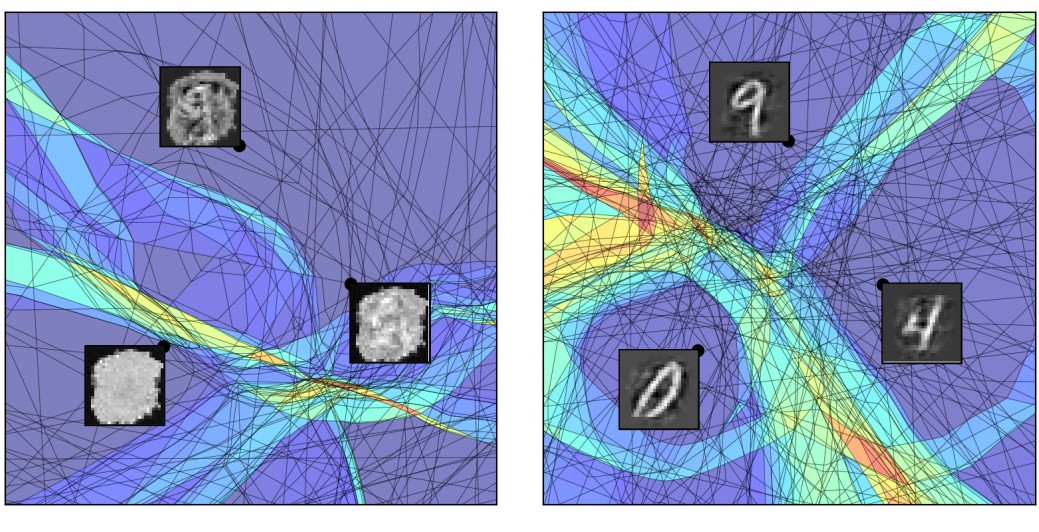

Centroids with large inner products        Aligned Centroids

Figure 10: Here we consider the centroids and functional geometry of the DNs of Figure 4 in the same way as Figure 1. On the **left** we consider the DN trained with weight decay and on the **right** we consider the DN trained with GrokAlign.

## H    Proofs of Main Results

**Proof of Theorem 2.**    Using Theorem 11 and Theorem 12, it follows that

$$
\begin{aligned}
\mu^{(1\leftarrow l)}_{\omega^{(1\leftarrow l)}_{\mathbf{x}}} &= \left(\mathbf{A}^{(l-1)}_{\omega^{(l-1)}_{\mathbf{x}}}\cdots\mathbf{A}^{(1)}_{\omega^{(1)}_{\mathbf{x}}}\right)^{\top}\mu^{(l)}_{\omega^{(l)}_{\mathbf{x}}}\\
&= \left(\mathbf{A}^{(l-1)}_{\omega^{(l-1)}_{\mathbf{x}}}\cdots\mathbf{A}^{(1)}_{\omega^{(1)}_{\mathbf{x}}}\right)^{\top}\left(\mathbf{A}^{(l)}_{\omega^{(l)}_{\mathbf{x}}}\right)^{\top}\mathbf{1}\\
&= \left(\mathbf{A}^{(l)}_{\omega^{(l)}_{\mathbf{x}}}\cdots\mathbf{A}^{(1)}_{\omega^{(1)}_{\mathbf{x}}}\right)^{\top}\mathbf{1}\\
&= \left(\mathbf{A}^{(1\leftarrow l)}_{\omega^{(1\leftarrow l)}_{\mathbf{x}}}\right)^{\top}\mathbf{1}.
\end{aligned}
$$

Extending this to the $L^{\text{th}}$ yields the desired result. $\qquad\square$

**Proof of Theorem 4.**    In a similar way to Proposition 9, one can show that

$$
\partial_t\mu_{\mathbf{x}} = \frac{\eta}{m}\sum_{p=1}^{m}\left(\left(m^{\top}_{\mathbf{x}_p}\mathbf{W}^{(2)}\mathbf{Q}\left[\mathbf{x}_p\right]\mathbf{Q}_{\mathbf{x}}\left(\mathbf{W}^{(2)}\right)^{\top}\mathbf{1}\right)\mathbf{x}_p\right.
$$

$$
\left. + \left(\mathbf{W}^{(1)}\right)^{\top}\mathbf{Q}_{\mathbf{x}}\sigma\left(\mathbf{W}^{(1)}\mathbf{x}_p\right)m^{\top}_{\mathbf{x}_p}\mathbf{1}\right).
$$

Using Lemma 14, this simplifies to

$$
\partial_t\left(\langle\mathbf{x}',\mu_{\boldsymbol{\nu}}\rangle\right) = \frac{\eta}{m}\sum_{p=1}^{m}\Theta\left(\mathbf{x}',\mathbf{x}_p\right)m_{\mathbf{x}_p}.
$$

$\qquad\square$

**Proof of Corollary 6.**    Using Theorem 2, the centroid of an aligned Jacobian is $\mu_{\mathbf{x}} = \mathbf{x}\mathbf{c}^{\top}\mathbf{1} = c\mathbf{x}$ where $c = \mathbf{c}^{\top}\mathbf{1}$. $\qquad\square$

**Proof of Theorem 7.**    Without loss of generality, we can assume $\mathbf{A}_{\omega_{\mathbf{x}}}$ to be of the form $\mathbf{c}\mathbf{v}^{\top}$ for some $\mathbf{v}\in\mathbb{R}^d$. In particular, we assume that the vector $\mathbf{v}$ is of the same norm as $\mathbf{x}$. Then, locally in $\omega_{\mathbf{x}}$, we have

$$
f(\mathbf{x}+\boldsymbol{\epsilon}) = \mathbf{c}\mathbf{v}^{\top}\left(\mathbf{x}+\boldsymbol{\epsilon}\right).
$$

Therefore, $\mathbf{x}$ will only be misclassified by the DN when $\mathbf{v}^{\top}\left(\mathbf{x}+\boldsymbol{\epsilon}\right) < 0$. From the Cauchy-Schwartz inequality we have that

$$
-\|\mathbf{v}\|_2\|\boldsymbol{\epsilon}\|_2 \le \mathbf{v}^{\top}\boldsymbol{\epsilon} < -\mathbf{v}^{\top}\mathbf{x}.
$$

Hence,

$$
\|\boldsymbol{\epsilon}\|_2 > \frac{\mathbf{v}^{\top}\mathbf{x}}{\|\mathbf{v}\|_2},
$$

the right-hand side of which is maximized when $\mathbf{v}$ is $\mathbf{x}$. $\qquad\square$

**Proof of Theorem 8.**    Since $\ell_p = \ell\left(f\left(\mathbf{x}_p\right)\right) = \ell\left(\mathbf{J}_{\mathbf{x}_p}+\mathbf{b}_{\mathbf{x}_p}\right)$, it follows from the chain rule that

$$
\frac{\partial\ell_p}{\partial\mathbf{j}^{(c)}_{\mathbf{x}_p}} = \frac{\partial\ell}{\partial\mathbf{j}^{(c)}_{\mathbf{x}_p}}\cdot\mathbf{x}_p.
$$

For simplicity we suppose that $\ell$ treats all wrong classes equally such that,

$$
\frac{\partial\ell_p}{\partial\mathbf{j}^{(c)}_{\mathbf{x}_p}} = \begin{cases}\beta_1(c)\mathbf{x}_p & c = y_p\\ \beta_2(c)\mathbf{x}_p & c \ne y_p,\end{cases}
\tag{2}
$$

however, the proof proceeds without this assumption too.

Note that the optimisation problem is convex on a convex set and thus it is sufficient to consider the Karush-Kuhn-Tucker conditions with a Lagrange multiplier. More specifically, since the Frobenius norm constraint implies that $\sum_{c=1}^{C} \left\| \mathbf{j}_{\mathbf{x}_p}^{(c)} \right\|_2^2 \leq \alpha$, we can consider

$$\ell_p^{(\text{KKT})} = \ell_p + \lambda \left( \sum_{c=1}^{C} \left\| \mathbf{j}_{\mathbf{x}_p}^{(c)} \right\|_2^2 - \alpha \right).$$

Thus,

$$\mathbf{0} = \frac{\partial \ell_p^{(\text{KKT})}}{\partial \mathbf{j}_{\mathbf{x}_p}^{(c)}} = \frac{\partial \ell_p}{\partial \mathbf{j}_{\mathbf{x}_p}^{(c)}} + 2\lambda \mathbf{j}_{\mathbf{x}_p}^{(c)} \tag{3}$$

for $c = 1, \ldots, C$, and

$$0 = \frac{\partial \ell^{(\text{KKT})}}{\partial \lambda} = \sum_{c=1}^{C} \left\| \mathbf{j}_{\mathbf{x}_p}^{(c)} \right\|_2^2 - \alpha. \tag{4}$$

From Equation (3), we have

$$0 = \sum_{c=1}^{C} \left\langle \mathbf{j}_{\mathbf{x}_p}^{(c)}, \frac{\partial \ell_p^{(\text{KKT})}}{\partial \mathbf{j}_{\mathbf{x}_p}^{(c)}} \right\rangle = \sum_{c=1}^{C} \left\langle \mathbf{j}_{\mathbf{x}_p}^{(c)}, \frac{\partial \ell_p}{\partial \mathbf{j}_{\mathbf{x}_p}^{(c)}} \right\rangle + 2\lambda \left\| \mathbf{j}_{\mathbf{x}_p}^{(c)} \right\|_2^2 \overset{Equation\ (4)}{=} 2\lambda\alpha + \sum_{c=1}^{C} \left\langle \mathbf{j}_{\mathbf{x}_p}^{(c)}, \frac{\partial \ell_p}{\partial \mathbf{j}_{\mathbf{x}_p}^{(c)}} \right\rangle,$$

which implies that $\lambda = -\frac{1}{2\alpha} \sum_{c=1}^{C} \left\langle \mathbf{j}_{\mathbf{x}_p}^{(c)}, \frac{\partial \ell_p}{\partial \mathbf{j}_{\mathbf{x}_p}^{(c)}} \right\rangle$. Using this back in Equation (3), we deduce that

$$\mathbf{0} = \frac{\partial \ell_p}{\partial \mathbf{j}_{\mathbf{x}_p}^{(c)}} - \frac{1}{\alpha} \sum_{c'=1}^{C} \left\langle \mathbf{j}_{\mathbf{x}_p}^{(c')}, \frac{\partial \ell_p}{\partial \mathbf{j}_{\mathbf{x}_p}^{(c')}} \right\rangle \mathbf{j}_{\mathbf{x}_p}^{(c)} \tag{5}$$

for $c = 1, \ldots, C$. Consider the ansatz

$$\mathbf{j}_{\mathbf{x}_p}^{(c)} = \begin{cases} a_1 \mathbf{x}_p & c = y_p \\ a_2 \mathbf{x}_p & c \neq y_p. \end{cases} \tag{6}$$

Then for $c = y_p$, Equation (5) becomes

$$\mathbf{0} = \beta_1 \mathbf{x}_p - \frac{1}{\alpha} \left( (C-1)a_2\beta_2 \left\| \mathbf{x}_p \right\|_2^2 + a_1\beta_1 \left\| \mathbf{x}_p \right\|_p^2 \right) a_1 \mathbf{x}_p$$

$$= \left( \alpha\beta_1 - ((C-1)a_2\beta_2 + a_1\beta_1)\, a_1 \left\| \mathbf{x}_p \right\|_2^2 \right) \mathbf{x}_p,$$

which implies that

$$0 = \alpha\beta_1 - \left( (C-1)a_1 a_2 \beta_2 + a_1^2 \beta_1 \right) \left\| \mathbf{x}_p \right\|_2^2. \tag{7}$$

Similarly, when $c \neq y_p$, from Equation (5) we deduce that

$$0 = \alpha\beta_2 - \left( (C-1)a_2^2 \beta_2 + a_1 a_2 \beta_1 \right) \left\| \mathbf{x}_p \right\|_2^2. \tag{8}$$

Furthermore, from Equation (4) we get

$$\alpha = \left\| \mathbf{x}_p \right\|_2^2 \left( a_1^2 + (C-1)a_2^2 \right). \tag{9}$$

Provided $\beta_1$ and $\beta_2$ are non-zero,

$$\begin{cases} a_1 = \frac{\beta_1 \sqrt{\alpha}}{\|\mathbf{x}_p\|_2 \sqrt{\beta_1^2 + (C-1)\beta_2^2}} \\ a_2 = -\frac{\beta_2 \sqrt{\alpha}}{\|\mathbf{x}_p\|_2^2 \sqrt{\beta_1^2 + (C-1)\beta_2^2}} \end{cases} \tag{10}$$

demonstrates that the systems of Equations (7) to (9) form a consistent system of equations that admit a unique solution. If $\beta_2$ equals zero, then

$$\begin{cases} a_1 = \frac{\sqrt{\alpha}}{\|\mathbf{x}_p\|_2} \\ a_2 = 0 \end{cases} \tag{11}$$

demonstrates that the systems of Equations (7) to (9) form a consistent system of equations that admit a unique solution. Therefore, $\mathbf{J}_{\mathbf{x}_p}$ as constructed in Equation (6) minimises the constrained optimization. $\qquad \square$

**Proof of Proposition 9.**   From Lemma 13, observe that

$$\partial_t \mu_{\mathbf{x}} = \left( \partial_t \left( \mathbf{W}^{(2)} \right) \mathbf{Q}_{\mathbf{x}} \mathbf{W}^{(1)} + \mathbf{W}^{(2)} \mathbf{Q}_{\mathbf{x}} \partial_t \left( \mathbf{W}^{(1)} \right) \right)^\top \mathbf{1},$$

where

$$\partial_t \left( \mathbf{W}^{(i)} \right) = -\eta \nabla_{\mathbf{W}^{(i)}} \mathcal{L}$$

for $i = 1, 2$. One can show that

$$\nabla_{\mathbf{W}^{(1)}} \mathcal{L} = -\frac{1}{m} \sum_{p=1}^m \left( \mathbf{W}^{(2)} \mathbf{Q}_{\mathbf{x}_p} \right)^\top \mathbf{m}_{\mathbf{x}_p} \mathbf{x}_p^\top$$

and

$$\nabla_{\mathbf{W}^{(2)}} \mathcal{L} = -\frac{1}{m} \sum_{p=1}^m \mathbf{m}_{\mathbf{x}_p} \sigma \left( \mathbf{W}^{(1)} \mathbf{x}_p \right)^\top.$$

Therefore,

$$\partial_t \mu_{\mathbf{x}} = \frac{\eta}{m} \sum_{p=1}^m \left( \left( \mathbf{m}_{\mathbf{x}_p}^\top \mathbf{W}^{(2)} \mathbf{Q}_{\mathbf{x}_p} \mathbf{Q}_{\mathbf{x}} \left( \mathbf{W}^{(2)} \right)^\top \mathbf{1} \right) \mathbf{x}_p \right.$$

$$\left. + \left( \mathbf{W}^{(1)} \right)^\top \mathbf{Q}_{\mathbf{x}} \sigma \left( \mathbf{W}^{(1)} \mathbf{x}_p \right) \mathbf{m}_{\mathbf{x}_p}^\top \mathbf{1} \right).$$

In particular,

$$\mathbf{m}_{\mathbf{x}_p}^\top \mathbf{1} = 1 - \frac{\sum_{c=1}^C \exp \left( [f_\theta \left( \mathbf{x}_p \right)]_c \right)}{\sum_{c'=1}^C \exp \left( [f_\theta \left( \mathbf{x}_p \right)]_{c'} \right)} = 0,$$

meaning

$$\partial_t \mu_{\mathbf{x}} = \frac{\eta}{m} \sum_{p=1}^m \left( \mathbf{m}_{\mathbf{x}_p}^\top \mathbf{W}^{(2)} \mathbf{Q}_{\mathbf{x}_p} \mathbf{Q}_{\mathbf{x}} \left( \mathbf{W}^{(2)} \right)^\top \mathbf{1} \right) \mathbf{x}_p.$$

Therefore, the result follows since $\partial_t \langle \mathbf{x}, \mu_{\mathbf{x}} \rangle = \langle \mathbf{x}, \partial_t \left( \mu_{\mathbf{x}} \right) \rangle$. $\qquad\square$

**Proof of Lemma 13.**   This follows immediately from the application of Theorem 2. $\qquad\square$

**Proof of Lemma 14.**   Observe that in this setting we have

$$\Theta \left( \mathbf{x}, \mathbf{x}' \right) = \langle \nabla_{\mathbf{W}^{(2)}} f_\theta(\mathbf{x}), \nabla_{\mathbf{W}^{(2)}} f_\theta \left( \mathbf{x}' \right) \rangle + \langle \nabla_{\mathbf{W}^{(1)}} f_\theta(\mathbf{x}), \nabla_{\mathbf{W}^{(1)}} f_\theta \left( \mathbf{x}' \right) \rangle.$$

Therefore, noting that

$$\nabla_{\mathbf{W}^{(2)}} f_\theta(\mathbf{x}) = \sigma \left( \mathbf{W}^{(1)} \mathbf{x} \right)$$

and

$$\nabla_{\mathbf{W}^{(1)}} f_\theta(\mathbf{x}) = \mathbf{W}^{(2)} \mathbf{Q}_{\mathbf{x}} \mathbf{x}^\top,$$

the result follows. $\qquad\square$

**Proof of Theorem 15.**   Using Theorem 11 and Theorem 12, it follows that

$$\tau_{\omega_{\mathbf{x}}^{(1 \leftarrow l)}}^{(1 \leftarrow l)} = \left\| \mu_{\omega_{\mathbf{x}}^{(l \leftarrow l)}}^{(1 \leftarrow l)} \right\|_2^2 + 2 \left( \mu_{\omega_{\mathbf{x}}^{(l)}}^{(l)} \right)^\top \mathbf{b}_{\omega_{\mathbf{x}}^{(1 \leftarrow l-1)}}^{(1 \leftarrow l-1)} + 2 \left( \mathbf{b}_{\omega_{\mathbf{x}}^{(l)}}^{(l)} \right)^\top \mathbf{1}$$

$$= \left\| \mu_{\omega_{\mathbf{x}}^{(l \leftarrow l)}}^{(1 \leftarrow l)} \right\|_2^2 + 2 \left( \left( \mathbf{A}_{\omega_{\mathbf{x}}^{(l)}}^{(l)} \right)^\top \mathbf{1} \right)^\top \mathbf{b}_{\omega_{\mathbf{x}}^{(1 \leftarrow l-1)}}^{(1 \leftarrow l-1)} + 2 \left( \mathbf{b}_{\omega_{\mathbf{x}}^{(l)}}^{(l)} \right)^\top \mathbf{1}$$

$$= \left\| \mu_{\omega_{\mathbf{x}}^{(l \leftarrow l)}}^{(1 \leftarrow l)} \right\|_2^2 + 2 \left( A_{\omega_{\mathbf{x}}^{(l)}}^{(l)} \mathbf{b}_{\omega_{\mathbf{x}}^{(l)}}^{(1 \leftarrow l-1)} + \mathbf{b}_{\omega_{\mathbf{x}}^{(l)}}^{(l)} \right)^\top \mathbf{1}$$

$$= \left\| \mu_{\omega_{\mathbf{x}}^{(l \leftarrow l)}}^{(1 \leftarrow l)} \right\|_2^2 + 2 \left( \mathbf{b}_{\omega_{\mathbf{x}}^{(l)}}^{(1 \leftarrow l)} \right)^\top \mathbf{1}.$$

Extending this to the $L^{\text{th}}$ yields the desired result. $\qquad\square$

# I SUPPORTING RESULTS

Note that, for a CPA DN $f = \left( f^{(L)} \circ \cdots \circ f^{(1)} \right)$, each $f^{(l)}$ and sub-component $f^{(1 \leftarrow l)} = \left( f^{(l)} \circ \cdots \circ f^{(1)} \right)$ are also CPA DNs. Let $\mathbf{A}^{(l)}_{\omega_{\mathbf{x}}^{(l)}}$, $\mathbf{b}^{(l)}_{\omega_{\mathbf{x}}^{(l)}}$, $\omega_{\mathbf{x}}^{(l)}$, $\mu^{(\ell)}_{\omega_{\mathbf{x}}^{(l)}}$ and $\mathbf{A}^{(1 \leftarrow l)}_{\omega_{\mathbf{x}}^{(1 \leftarrow l)}}$, $\mathbf{b}^{(1 \leftarrow l)}_{\omega_{\mathbf{x}}^{(1 \leftarrow l)}}$, $\omega_{\mathbf{x}}^{(1 \leftarrow l)}$, $\mu^{(1 \leftarrow l)}_{\omega_{\mathbf{x}}^{(1 \leftarrow \ell)}}$ be analogous notation for the layer and sub-component networks to that of the CPA DNs we introduced in Section 2.1.

**Theorem 11** (Balestriero et al. 2019). *The $l^{th}$ layer of a DN partitions its input space according to a power diagram with centroids*

$$\mu^{(l)}_{\omega_{\mathbf{x}}^{(l)}} = \left( \mathbf{A}^{(l)}_{\omega_{\mathbf{x}}^{(l)}} \right)^{\top} \mathbf{1},$$

*and radii*

$$\tau^{(\ell)}_{\omega_{\mathbf{x}}^{(l)}} = \left\| \mu^{(l)}_{\omega_{\mathbf{x}}^{(l)}} \right\|_2^2 + 2 \left( \mathbf{b}_{\omega_{\mathbf{x}}^{(l)}} \right)^{\top} \mathbf{1}.$$

**Theorem 12** (Balestriero et al. 2019). *The continuous piecewise operation of a DN from the input to the output of the $l^{th}$ layer partitions its input space according to a power diagram with centroids*

$$\mu^{(1 \leftarrow l)}_{\omega_{\mathbf{x}}^{(1 \leftarrow \ell)}} = \left( \mathbf{A}^{(l-1)}_{\omega_{\mathbf{x}}^{(l-1)}} \cdots \mathbf{A}^{(1)}_{\omega_{\mathbf{x}}^{(1)}} \right)^{\top} \mu^{(l)}_{\omega_{\mathbf{x}}^{(l)}} =: \left( \mathbf{A}^{(1 \leftarrow l-1)}_{\omega_{\mathbf{x}}^{(1 \leftarrow l-1)}} \right)^{\top} \mu^{(l)}_{\omega_{\mathbf{x}}^{(l)}}$$

*and radii*

$$\tau^{(1 \leftarrow l)}_{\omega_{\mathbf{x}}^{(1 \leftarrow l)}} = \left\| \mu^{(1 \leftarrow l)}_{\omega_{\mathbf{x}}^{(1 \leftarrow l)}} \right\|_2^2 + 2 \left( \mu^{(l)}_{\omega_{\mathbf{x}}^{(l)}} \right)^{\top} \mathbf{b}^{(1 \leftarrow l-1)}_{\omega_{\mathbf{x}}^{(1 \leftarrow l-1)}} + 2 \left( \mathbf{b}^{(l)}_{\omega_{\mathbf{x}}^{(l)}} \right)^{\top} \mathbf{1}$$

**Lemma 13.** *In the setting of Theorem 4, we have $\mu_{\mathbf{x}} = \left( \mathbf{W}^{(2)} \mathbf{Q}_{\mathbf{x}} \mathbf{W}^{(1)} \right)^{\top} \mathbf{1}$, where $\mathbf{Q}_{\mathbf{x}} := \text{diag} \left( \sigma' \left( \mathbf{W}^{(1)} \mathbf{x} \right) \right)$. (Proof in Appendix H).*

**Lemma 14.** *In the setting of Theorem 4, the neural tangent kernel of $f_\theta$ between $\mathbf{x}, \mathbf{x}' \in \mathbb{R}^d$ is given by*

$$\Theta \left( \mathbf{x}, \mathbf{x}' \right) = \sigma \left( \mathbf{W}^{(1)} \mathbf{x} \right)^{\top} \sigma \left( \mathbf{W}^{(1)} \mathbf{x}' \right) + \left( \mathbf{x}^{\top} \mathbf{x}' \right) \left( \mathbf{W}^{(2)} \mathbf{Q}_{\mathbf{x}} \mathbf{Q}_{\mathbf{x}'} \left( \mathbf{W}^{(2)} \right)^{\top} \right).$$

*(Proof in Appendix H).*

**Theorem 15.** *For a DN $f$, we have $\tau_{\mathbf{x}} = \| \mu_{\mathbf{x}} \|_2^2 - 2 \left( \mathbf{b}_{\mathbf{x}} \right)^{\top} \mathbf{1}$. (Proof in Appendix H).*

