# OpenReview forum: "What Deep Networks Want to Learn and How to Get There"
_ICLR.cc/2026/Conference — Submitted to ICLR 2026_

### Official Review · Reviewer_3JCQ · 2025-10-31

**Soundness:** 2
**Presentation:** 2
**Contribution:** 2
**Rating:** 2
**Confidence:** 3

**Summary:**

This paper studies a theoretical framework to explain the generalization and robustness properties of neural networks. The authors introduce two concepts : *centroid alignment* and *Jacobian alignment*. They argue that the first is connected to generalization, while the second favors robustness.

The starting point is that typical neural networks $f$ define piecewise affine functions. Following Balestriero et al. (2019), the input space can be partitioned in regions $\omega_\nu$ on which $f$ is locally linear. This partition forms a power diagram (a generalized Voronoi diagram) with centroids $\mu_\nu$. A network is said to be **centroid-aligned** at point $x$ if the centroid $\mu_x$ associated to the region containing $x$ is proportional to $x$. The authors provide heuristic arguments to claim that this property promotes better generalization (Theorem 4).

Next, the authors define **Jacobian alignment** at $x$ by the property $J_x(f) = cx^T$, where $J_x(f)$ is the Jacobian of the network at $x$. They argue (Theorem 7) that this condition promotes robustness of the network.

Finally, the authors perform empirical experiments to validate their theory. On the XOR classification task, they measure the centroid alignment and show that it is correlated with the test loss.

They also propose a new method, called GrokAlign, which aims at promoting Jacobian alignment by penalizing the Jacobian norm in the training loss, in order to accelerate the "grokking" phenomenon (delayed improvement in test loss). They compare this methods with other algorithms that pursue the same goal, on several classification tasks.

**Strengths:**

- The paper addresses an central topic in machine learning (generalization and robustness of neural networks), whose importance is well motivated, with an original approach,
- The authors perform extensive and detailed empirical experiments to validate their theory

**Weaknesses:**

While the proposed idea is interesting and original, the theoretical framework, in its current form, lacks sufficient soudndess to be convincing. The analysis would benefit from more rigour and clarity.

- in Section 2.2, the authors claim that centroid alignment promotes generalization. However, this argument developed in lines 187-199 is quite heuristic and imprecise. The whole reasoning relies of the fact that "$\Theta$ (the neural tangent kernel) and $m$ are uncorrelated". However, I do not see why this is the case: they both depend on the neural network parameters. I believe that there is an interesting idea to be explored there, but the analysis should be developed much more thoroughly. Therefore, stating that "[the authors have] established that centroid-aligned DNs exhibit generalization" (line 203) seems to be a strong overclaim.

- in Section 2.3, the authors introduced Jacobian alignment as $J_x(f) = cx^T$. This seems excessively restrictive: this means that locally around $x$, the network output is invariant in *any* direction orthogonal to $x$. Are there examples of such situation, even in simple cases?

- Theorem 7, which supports the idea that Jacobian alignment promotes generalization, is also quite confusing and heuristic. Again, there appears to be interesting insight, but the statement (and the proof) lack clarity in exposition. The authors could address it by illustrating the claim on a simple toy problem.

- The organization of the appendix is confusing: for instance, Proposition 9 uses the $Q_x$ notation, which is only introduced in Lemma 13, at the very end. This makes the proofs hard to follow.

Finally, although empirical experiments are quite extensive, they only limited support for the theory. While figure 2 indicates that centroid alignment increases during training, it stays at a relatively low value. Besides, this is only for one point in the dataset; how is alignment defined *globally*? (see Q1 below)

**Questions:**

1. What does it mean for a network to be centroid (or Jacobian) aligned? Indeed, the authors have defined those properties for a given input $x$. It is clear that this cannot hold for any $x$ (since $\mu_x$ is locally constant, it cannot be equal to $x$, unless the power diagram is degenerate). Do the authors mean that this property holds for every *training* data point (in which case it implies that every data point must lie in a different region of the diagram)?

2. In the proof of Theorem 7, the authors state that "x will only be missclassified by the DN when $v^T(x_\epsilon) < 0$" (l. 1062): why is this the case? There seems to be no indication on the related classification task.

3. The proof of Theorem 8 starts with the statement (l. 1071): $l(f(x_p)) = l(J_{x_p} + b_{x_p})$. Since $J_{x_p}$ is the Jacobian and $b_{x_p}$ the bias vector, those have different dimensions and cannot be added. If I understand correctly, we should have locally $f(x) = J_{x_p} x + b_{x_p}$. Is there maybe a term in $x$ missing?

---

> ### Author Response · Authors · 2025-11-20
>
> Thank you for your detailed review of our paper that addresses a central problem in machine learning with an approach that is well-motivated. We also appreciate you acknowledging our extensive empirical experiments that validate our theory.

---

> ### Author Response · Authors · 2025-11-20
> **Concern Regarding Connection between Centroid Alignment and Generalisation**
>
> **in Section 2.2, the authors claim that centroid alignment promotes generalization. However, this argument developed in lines 187-199 is quite heuristic and imprecise.**
>
> We will make this link clearer and more rigorous in our updated paper. We have both theoretical and empirical rationales for this link. Theoretically, in our submitted paper, we have analyzed a two-layer, scalar-output deep network (DN) and rigorously proved that a changing alignment between the centroids and the training data corresponds to changing values in the neural tangent kernel evaluated at the training data. Prior work (Woodworth et al., 2020; Moroshko et al., 2020) has connected the period of training where the neural tangent kernel changes to the feature learning regime of training. Additional prior work (Baratin et al., 2020, Fort et al., 2020, Paccolat et al., 2020) has empirically connected the feature learning regime of training to improved generalization. Furthermore, prior work cited in our submitted paper (Lyu et al., 2024; Rubin et al., 2024; Kumar et al., 2024) makes the connection between the feature learning regime and delayed generalization (aka, grokking). These papers show empirically that DNs that have commenced feature learning exhibit greater generalisation. We plan to update our paper with the arguments above, and so would appreciate your comments. Nevertheless, in the absence of a direct rigorous theoretical result connecting centroid alignment to generalisation, in the updated paper, we will tone down our claims appropriately.
>
> Baratin, Aristide, et al. "Implicit regularization via neural feature alignment." International Conference on Artificial Intelligence and Statistics. PMLR, 2021.
>
> Fort, Stanislav, et al. "Deep learning versus kernel learning: an empirical study of loss landscape geometry and the time evolution of the neural tangent kernel." Advances in Neural Information Processing Systems 33 (2020): 5850-5861.
>
> Paccolat, Jonas, et al. "Geometric compression of invariant manifolds in neural networks." Journal of Statistical Mechanics: Theory and Experiment 2021.4 (2021): 044001.
>
> **The whole reasoning relies of the fact that "$\Theta$ (the neural tangent kernel) and $m$ are uncorrelated". However, I do not see why this is the case: they both depend on the neural network parameters. I believe that there is an interesting idea to be explored there, but the analysis should be developed much more thoroughly.**
>
> We will improve the text in lines 187-199 along the lines of the argument in our response. We agree with you that this is an interesting area for exploration, and upon review we believe the connection to be more nuanced than we presented in the paper. To disentangle $\boldsymbol{\Theta}$ and $\mathbf{m}$, in our updated paper, we will introduce the assumption that the DN has been trained sufficiently to have memorised/interpolated the training data (i.e., the training error is small) so that $\mathbf{m}$ can be treated as constant. In this setting, we can rigorously prove that a DN that is becoming increasingly centroid alignment is equivalent to the DN being in the feature learning regime of training.

---

> ### Author Response · Authors · 2025-11-20
> **Concerns Regarding the Property of Jacobian Alignment**
>
> **in Section 2.3, the authors introduced Jacobian alignment as $J_x(f) = cx^T$. This seems excessively restrictive: this means that locally around $x$, the network output is invariant in any direction orthogonal to $x$. Are there examples of such situation, even in simple cases?**
>
> We are sorry for any confusion caused by our argument in Section 2.3. We introduce the concept of Jacobian alignment in Definition 5. We then prove in Theorem 8 that an aligned Jacobian is the solution at the global optimum of a wide range of different loss functions (including negative cross entropy and squared error) featuring a wide range of different penalty terms. We also prove in Theorem 7 that an aligned Jacobian produces a DN with optimal robustness properties. Consequently, a rank-1, aligned Jacobian can be regarded as the “ultimate destination” of deep net training. We can understand your surprise, because we were surprised, too, when we discovered this result. For it has profound consequences. Indeed, the fact that the network output is invariant in any direction orthogonal to the data point x can be motivated and explained through the lens of optimal “matched filters” from signal and image processing (Balestriero and Baraniuk 2021). For additional discussion, please see the discussion regarding “The theoretical framework seems to hinge on the assumption that the input-output Jacobian converges to rank-1 matrix…” with Reviewer PBfN.
>
> R. Balestriero and R. G. Baraniuk, "Mad Max: Affine Spline Insights Into Deep Learning," in Proceedings of the IEEE, vol. 109, no. 5, pp. 704-727, May 2021, doi: 10.1109/JPROC.2020.3042100
>
> **Theorem 7, which supports the idea that Jacobian alignment promotes generalization, is also quite confusing and heuristic. Again, there appears to be interesting insight, but the statement (and the proof) lack clarity in exposition. The authors could address it by illustrating the claim on a simple toy problem.**
>
> Theorem 7 is a result about the robustness of Jacobian alignment, rather than a statement about generalization. The statement and proof of Theorem 7 will be clarified in an updated version of the paper. Please see our response below for more details. To illustrate how Jacobian alignment corresponds to an increase in robustness, we will add a line to Figure 4 showing how Jacobian alignment evolves over the course of training.

---

> ### Author Response · Authors · 2025-11-20
> **Concerns Regarding the Centroid Alignment Property**
>
> **Finally, although empirical experiments are quite extensive, they only limited support for the theory. While figure 2 indicates that centroid alignment increases during training, it stays at a relatively low value. Besides, this is only for one point in the dataset; how is alignment defined globally? (see Q1 below)**
>
> While the peak centroid alignment value in Figure 2 is only about $0.175$, given the high-dimensionality of the input data space, this value is statistically very significant.
>
> For an input point in d dimensions, we can consider sampling a direction uniformly at random. The cosine similarity between this direction and the point follows a probability distribution. For large $d$, this is well approximated by a normal distribution with zero mean and standard deviation $\frac{1}{\sqrt{d}}$. Therefore, we can construct an interval of the form $(-c,c)$ which encompasses approximately $95$% of the probability mass of this distribution. In Figure 2, the input dimension is $40,000$, thus $c$ is approximately equal to $0.0098$. Therefore, an alignment of $0.175$ is very significant. We will add a clarification about this calibration both in the main text and the caption of Figure 2 in the revised paper. We will include the details for the calibration in a new appendix.
>
> You are correct that Figure 2 considers alignment at one point in the training dataset. However, all of the rest of the results in the paper are computed at all points in the entire training dataset. See, for example, the blue lines in Figure 4 and red lines in Figure 5.
>
> **What does it mean for a network to be centroid (or Jacobian) aligned?...**
>
> We really appreciate your comment because it has pointed to a key point that we will clarify in our updated paper. For centroid alignment, we will make it clear that Definition 3 deals with alignment at a single point $\mathbf{x}$. We will then introduce a new definition of a “Centroid-aligned DN” in which we have alignment at each and every point in the training dataset. We will do the same for Jacobian alignment. You are correct that every training data point must lie in a different linear region of the power diagram. Fortunately, this is easy to satisfy in practice. Since the number of linear regions grows exponentially with the number of neurons (Boris and Rolnick, 2019), in practice, the linear regions are very small and localized.
>
> Hanin, Boris, and David Rolnick. "Complexity of linear regions in deep networks." International Conference on Machine Learning. PMLR, 2019.

---

> ### Author Response · Authors · 2025-11-20
> **Clarity in Mathematical Results**
>
> **The organization of the appendix is confusing: for instance, Proposition 9 uses the $Q_x$ notation, which is only introduced in Lemma 13, at the very end. This makes the proofs hard to follow.**
>
> We appreciate your comment. We will carefully revise the paper to make the mathematical arguments easier to follow, including re-ordering the appendices to match the development of the ideas in the main text, and introducing all notation at the beginning of each appendix.
>
> **In the proof of Theorem 7, the authors state that "x will only be missclassified by the DN when $v^T(x_\epsilon) < 0$" (l. 1062): why is this the case? There seems to be no indication on the related classification task.**
>
> Thank you for raising this concern. In an updated version of the paper, we will clarify this step. The updated proof will proceed as follows.
>
> Without loss of generality, we can assume $\mathbf{A}\_{\omega_{\mathbf{x}}}$  to be of the form $\mathbf{c}\mathbf{v}^\top$ for some $\mathbf{v}\in\mathbb{R}^d$. In particular, we assume that the vector $\mathbf{v}$ is of the same norm as $\mathbf{x}$. Then, locally in $\omega\_{\mathbf{x}}$, we have $$f(\mathbf{x}+\boldsymbol{\epsilon})=\mathbf{c}\mathbf{v}^\top\left(\mathbf{x}+\boldsymbol{\epsilon}\right).$$ Without loss of generality, we can suppose that $\mathbf{v}^\top\mathbf{x}>0$. Otherwise, one can just let $\tilde{\mathbf{c}}=-\mathbf{c}$, $\tilde{\mathbf{v}}=-\mathbf{v}$ and consider $\mathbf{A}\_{\omega_{\mathbf{x}}}$ to be of the form $\tilde{\mathbf{c}}\tilde{\mathbf{v}}^\top$. The predicted class of $\mathbf{x}$ by the DN is given by $\arg\max\_i\left(\mathbf{c}\_i\mathbf{v}^\top\mathbf{x}\right)=\arg\max\_i\left(\mathbf{c}_i\right)$. Thus, for $\mathbf{x}$ to be misclassified, it must be the case that $\arg\max\_i\left(\mathbf{c}\_i\mathbf{v}^\top(\mathbf{x}+\boldsymbol{\epsilon})\right)\neq\arg\max\_i\left(\mathbf{c}_i\right)$, which can only happen if $\mathbf{v}^\top\left(\mathbf{x}+\boldsymbol{\epsilon}\right)<0$.
> From the Cauchy-Schwartz inequality we have that $$-\Vert\mathbf{v}\Vert\_2\Vert\boldsymbol{\epsilon}\Vert\_2\leq\mathbf{v}^\top\boldsymbol{\epsilon} < -\mathbf{v}^\top\mathbf{x}.$$Hence, $$\Vert\boldsymbol{\epsilon}\Vert_2 > \frac{\mathbf{v}^\top\mathbf{x}}{\Vert\mathbf{v}\Vert\_2},$$the right-hand side of which is maximized when $\mathbf{v}$ is $\mathbf{x}$.
>
> This updated proof provides extra details on the step of the proof you referenced, which helps improve the clarity and rigour of the proof.
>
> **The proof of Theorem 8 starts with the statement (l. 1071): $l(f(x_p)) = l(J_{x_p} + b_{x_p})$. Since $J_{x_p}$ is the Jacobian and $b_{x_p}$ the bias vector, those have different dimensions and cannot be added. If I understand correctly, we should have locally $f(x) = J_{x_p} x + b_{x_p}$. Is there maybe a term in $x$ missing?**
>
> Thank you for pointing this out. In our revised paper, we will add the $\mathbf{x}$ that is missing on line 1071.

---

### Official Review · Reviewer_Kxzz · 2025-10-31

**Soundness:** 3
**Presentation:** 2
**Contribution:** 2
**Rating:** 2
**Confidence:** 3

**Summary:**

The authors study the dynamics of deep neural networks through the lens of "functional geometry", defined as the partition of input space into regions where the network output is linear or approximately linear. Using this tool, they claim a connection between the emergence of generalisation and the simplification of the functional geometry: during training, the linear regions containing training data expand, while those around decision boundaries contract. This behaviour is quantified via "centroid alignment", a measure of how the network’s representation geometry evolves. Furthermore, the authors claim that centroid alignment is followed by "Jacobian alignment", where the network output becomes robust to local perturbations. The theoretical framework is supported by experiments on shallow networks trained on simple tasks such as XOR and MNIST. Finally, the authors introduce *GrokAlign*, a novel regularisation scheme that promotes convergence toward Jacobian-aligned solutions.

**Strengths:**

* The training dynamics of deep neural networks remain an open problem, and it is valuable to approach it with new perspectives and frameworks such as functional geometry.
* The paper is clearly motivated: the authors start from a precise conceptual goal and provide intuitive visualisations (e.g. Figure 1) that help the reader grasp the setting;
* The approach of using the developed theoretical intuition to design a new algorithm is commendable, as it both tests the proposed framework and strengthens the connection to practical applications.

**Weaknesses:**

1. The claimed connection between generalisation and simplification of the functional geometry is never theoretically supported. Is Theorem 4 intended to provide such support? If so, I disagree: showing that the NTK must change to reach alignment does not imply anything about generalisation.
2. The empirical validation is weak. In Figure 3, there is no significant difference between the curves for weight decay and those for grokalign; in Figure 4, the test and adversarial accuracies achieved by the two methods are comparable. In addition, I believe that a paper having deep networks in the title should at least consider three-layer networks (two-layer nets are still considered shallow from many points of view), and many datasets beyond the very simple MNIST are easily accessible with modern computational resources.
3. The presentation of some mathematical results is confusing. As mentioned above, the connection with generalisation is never clearly established. The proof of Theorem 4 is difficult to follow (Appendix H introduces a new undefined variable Q and references other unlisted propositions and lemmas), and the argument at the end of Section 2.2 is vague---what does it mean that alignment ensures that the linear regions of the partition maintain a coherent structure with the overall semantics?

---

**Questions:**

Despite the interesting approach, the weaknesses above make it impossible for me to recommend acceptance. It is possible that some might stem from a misunderstanding: in any case, I would raise my score if the weaknesses (1 and 2 in particular) are addressed satisfactorily.

---

> ### Author Response · Authors · 2025-11-20
>
> Thank you for your detailed review of our paper and for acknowledging the value of our approach and new perspectives. We also appreciate your support for the clarity of our exposition and our theory-driven approach.

---

> ### Author Response · Authors · 2025-11-20
> **Concerns Regarding the Connection Between Generalisation and the Functional Geometry of a DN**
>
> **“The claimed connection between generalisation and simplification of the functional geometry is never theoretically supported”**
>
> We will make this link clearer and more rigorous in our updated paper. We have both theoretical and empirical rationales for this link. Theoretically, in our submitted paper, we have analyzed a two-layer, scalar-output deep network (DN) and rigorously proved that a changing alignment between the centroids and the training data corresponds to changing values in the neural tangent kernel evaluated at the training data. Prior work (Woodworth et al., 2020; Moroshko et al., 2020) has connected the period of training where the neural tangent kernel changes to the feature learning regime of training. Additional prior work (Baratin et al., 2020, Fort et al., 2020, Paccolat et al., 2020) has empirically connected the feature learning regime of training to improved generalization. Furthermore, prior work cited in our submitted paper (Lyu et al., 2024; Rubin et al., 2024; Kumar et al., 2024) makes the connection between the feature learning regime and delayed generalization (aka, grokking). These papers show empirically that DNs that have commenced feature learning exhibit greater generalisation. We plan to update our paper with the arguments above, and so would appreciate your comments. Nevertheless, in the absence of a direct rigorous theoretical result connecting centroid alignment to generalisation, in the updated paper, we will tone down our claims appropriately.
>
> Baratin, Aristide, et al. "Implicit regularization via neural feature alignment." International Conference on Artificial Intelligence and Statistics. PMLR, 2021.
>
> Fort, Stanislav, et al. "Deep learning versus kernel learning: an empirical study of loss landscape geometry and the time evolution of the neural tangent kernel." Advances in Neural Information Processing Systems 33 (2020): 5850-5861.
>
> Paccolat, Jonas, et al. "Geometric compression of invariant manifolds in neural networks." Journal of Statistical Mechanics: Theory and Experiment 2021.4 (2021): 044001.
>
> **“I disagree: showing that the NTK must change to reach alignment does not imply anything about generalisation.”**
>
> Figure 4 in our submitted paper showed that, for a three hidden-layer, fully connected DN trained on MNIST, as centroid alignment (which captures the simplification in the function geometry) increases, the test accuracy also increases. In our updated paper, we plan to update Figure 4 to directly show the positive correlation between centroid alignment and test accuracy by plotting their values as a scatter plot. This figure will provide even stronger empirical support for our claim that when a DN is becoming centroid aligned, it starts to exhibit generalisation.

---

> ### Author Response · Authors · 2025-11-20
> **Concerns Regarding the Empirical Validation, and Mathematical Presentation**
>
> **“The empirical validation is weak. In Figure 3, there is no significant difference between the curves for weight decay and those for grokalign; in Figure 4, the test and adversarial accuracies achieved by the two methods are comparable. In addition, I believe that a paper having deep networks in the title should at least consider three-layer networks (two-layer nets are still considered shallow from many points of view), and many datasets beyond the very simple MNIST are easily accessible with modern computational resources.”**
>
> We apologize for the confusion around Figure 3. In fact, the experiment reported in Figure 3 was intended to highlight that, in practice, a range of different regularizations (in this case weight-decay and Jacobian Frobenius norm) will decrease the effective rank of the Jacobian over training iterations. We plan to replace this figure with a more insightful and less confusing example in the revised paper.
> In Figure 4, while the test and adversarial accuracies obtained by the two methods are comparable, the speeds at which those accuracies are achieved are very different. In particular (see the caption of Figure 4, which we will make more clear in the revised paper), the DN trained by GrokAlign achieves the level of $60$% robust accuracy in $62$% fewer epochs than the DN trained with weight-decay.  Another key takeaway from Figure 4 is that GrokAlign significantly improves the centroid alignment of the DN as compared to weigh-decay (from $0.25$ to $0.85$, which is substantial given the high dimensionality of the input data space). We can make this claim rigorous as follows:
> For an input point in $d$ dimensions, we can consider sampling a direction uniformly at random. The cosine similarity between this direction and the point follows a probability distribution. For large $d$, this is well approximated by a normal distribution with zero mean and standard deviation $\frac{1}{\sqrt{d}}$. Therefore, we can construct an interval of the form $(-c,c)$ which encompasses approximately $95$% of the probability mass of this distribution. In Figure 4, the input dimension is $784$, thus c is approximately equal to $0.07$. Therefore, an alignment of $0.85$ is very significant. We will add a clarification about this calibration both in the main text and the caption of Figure 4 in the revised paper. We will include the details for the calibration in a new appendix.
>
> We agree that we should illustrate our results with deeper than two-hidden-layer networks. In fact, the experiment reported in Figure 4 actually uses DNs with three hidden layers; please see the experimental details in Appendix A.3. We are sorry for not making this more clear; we will clarify appropriately in the revised paper. Furthermore, Figure 5 considers a 7-layer DN comprising 5 convolutional layers followed by 2 fully connected layers. This DN is also trained on CIFAR10, which is more realistic than MNIST.
>
>
> **“The presentation of some mathematical results is confusing. As mentioned above, the connection with generalisation is never clearly established. The proof of Theorem 4 is difficult to follow (Appendix H introduces a new undefined variable Q and references other unlisted propositions and lemmas), and the argument at the end of Section 2.2 is vague---what does it mean that alignment ensures that the linear regions of the partition maintain a coherent structure with the overall semantics?”**
>
> We will significantly update the proof of Theorem 4 and the argument at the end of Section 2.2 to make the connection between generalisation and centroid alignment rigorous; this will involve re-ordering Appendices F and H and adding some additional lemmas. We will also remove jargon such as “coherent structure” and “semantics” from the end of Section 2.2 and replace them with clear, direct arguments.

---

> > ### Comment · Reviewer_Kxzz · 2025-11-25
> >
> > Thank you for your response. After reading it, the weaknesses 1 and 2 I identified still stand, therefore my rating remains.
> >
> > Concerning 1 (which was also listed by 2 other reviewers), the authors themselves acknowledge that their theory simply shows that alignment implies changing the kernel. While changing the kernel (i.e. feature learning) often results in better generalisation, there are many counterexamples. As a result, there are no theoretical implications for generalisation coming from this analysis.
> >
> > Concerning 2, I still think that the differences in performance are minimal and there is no systematic exploration, or at least not enough to support the claims.

---

### Official Review · Reviewer_4vUT · 2025-11-01

**Soundness:** 3
**Presentation:** 2
**Contribution:** 3
**Rating:** 6
**Confidence:** 3

**Summary:**

The paper argues that SGD training in deep networks is headed toward a geometric 'destination' called 'Jacobian alignment'. First, networks generalize once training examples become maximally cosine‑aligned with the centroid of the piecewise‑linear region that contains them (centroid alignment).

Then, with more training, networks become robust when each row of the input‑output Jacobian at a training point is aligned with that point (Jacobian alignment).

The authors also propose GrokAlign, a Jacobian‑norm regularizer that pushes networks directly toward this destination and thereby accelerates grokking and improves robustness.

**Strengths:**

Well motivated with the toy setup of the centroids: They start from a concrete geometric model, each region has a centroid equal to the sum of Jacobian rows, so alignment becomes both interpretable and easy to measure.

Condensing all the ideas into the simple notion of Jacobian alignment make it easy to understand: The paper collapses multiple phenomena into one geometric condition: Jacobian alignment (each Jacobian row at x is parallel to x). This single notion subsumes centroid alignment and yields a clean understanding of how robustness emerges.

Results that support using GrokAlign: GrokAlign adds an average Jacobian Frobenius‑norm penalty to enforce the constraint in Theorem 8, which proves that bounded‑norm Jacobians drive the optimizer toward aligned solutions. Empirically, GrokAlign accelerates grokking and raises alignment: on MNIST it achieves 60% adversarial accuracy in 62% fewer epochs than weight decay.

**Weaknesses:**

While the core idea is compelling, the exposition could be clearer. The abstract is long and dense. It could be better understood at a higher level than describing Jacobian and centroid alignment. It’s hard to use as a stand‑alone summary of what the paper claims. For instance, a potential framing could be: "Deep nets divide the input space into locally linear regions; during training, generalization occurs when each example aligns with the centroid of its region, and robustness arrives when the network’s input‑sensitivity (i.e. Jacobian) also points in that same direction."

A simplified version of Figure 1 with a conceptual version of the centroid alignment rather than using an actual input-space partition would help substantially.

Section 2 then dives straight into heavy formalism (functional geometry, NTK, multiple definitions and theorems) before offering a plain‑language overview or running example, which makes the pitch hard to penetrate for readers without a theory background.

**Questions:**

Is the l2 adversarial robustness tied to the Frobenius norm of the Jacobian? If there were other types of adversarial perturbations, would a different norm over the Jacobian be required for GrokAlign?

"One limitation of our investigation is the dependency on the assumption that gradient descent has an implicit bias to minimise the rank of the weight matrices of the DN."
https://arxiv.org/pdf/2103.10427 This work presents empirical evidence that deep nets trained with gradient descent tend to end up with low‑rank representations, and in the linear case this directly means low‑rank weights.

---

> ### Author Response · Authors · 2025-11-20
>
> Thank you for your careful review of our paper and for agreeing that centroid and Jacobian alignment are concise notions to describe how robustness emerges in deep network (DN) training. We also appreciate your support for GrokAlign as a practical proof-of-concept for our theoretical findings.

---

> ### Author Response · Authors · 2025-11-20
> **Concerns Regarding Clarity and Extension to Other Adversarial Norms**
>
> **While the core idea is compelling, the exposition could be clearer. The abstract is long and dense. It could be better understood at a higher level than describing Jacobian and centroid alignment. It’s hard to use as a stand‑alone summary of what the paper claims. For instance, a potential framing could be: "Deep nets divide the input space into locally linear regions; during training, generalization occurs when each example aligns with the centroid of its region, and robustness arrives when the network’s input‑sensitivity (i.e. Jacobian) also points in that same direction."**
>
> Regarding the exposition and abstract, we appreciate your comments. In the updated paper, we plan to update the abstract and introduction so that they comprise a higher-level "executive summary’" of the entire paper that introduces a simple running example that we can elaborate on as we layer in theoretical and empirical results in the later sections.
>
> **A simplified version of Figure 1 with a conceptual version of the centroid alignment rather than using an actual input-space partition would help substantially.**
>
> Regarding Figure 1, this is a good idea, and we will plan to add additional lower-dimensional examples to the introduction and an appendix to drive home the concept of centroid and Jacobian alignment.
>
> **Section 2 then dives straight into heavy formalism (functional geometry, NTK, multiple definitions and theorems) before offering a plain‑language overview or running example, which makes the pitch hard to penetrate for readers without a theory background.**
>
> Regarding Section 2, we feel that updating the abstract and introduction so that they comprise a higher-level ``executive summary’’ of the entire paper (as we suggest just above) will make our theoretical and empirical discussions in Sections 2 onward more accessible to the reader.
>
> **Is the l2 adversarial robustness tied to the Frobenius norm of the Jacobian? If there were other types of adversarial perturbations, would a different norm over the Jacobian be required for GrokAlign?**
>
> This is an interesting question. Actually, the l2 adversarial robustness in Theorem 7 follows from the fact that we are considering rank one Jacobians, which have the form cx^T for vectors c and x. Because of this special rank-1 structure, the norm of such a matrix under the Frobenius, Nuclear, or Spectral norm is simply equivalent to the product of the l2 norms of the vectors c and x. Our robustness proof follows directly from this fact.
> The situation will be different for other norms, which is an interesting avenue for future work. For example, matrix 1-norm and Infinity-norm of a rank-1 matrix also decouples into a product of the norms of c and x; however, the norms involved in both cases are the 1- and Infinity vector norms.

---

> ### Author Response · Authors · 2025-11-20
> **Supporting the Low-Rank Assumption on the Jacobians of DNs**
>
> **“One limitation of our investigation is the dependency on the assumption that gradient descent has an implict bias to minimise the rank of the weight matrices of the DN” https://arxiv.org/pdf/2103.10427 This work presents empirical evidence that deep nets trained with gradient descent tend to end up with low‑rank representations, and in the linear case this directly means low‑rank weights.**
>
> Thank you for pointing out the paper https://arxiv.org/pdf/2103.10427. We will emphasize it in our updated paper because it bolsters our theoretical and empirical results that the Jacobians of even large-scale, state-of-the-art DNs are very low rank.
>
> In an updated version of the paper, our theoretical and empirical results on the low rankedness of DN Jacobians will proceed as follows:
>
> Our main theoretical result in Theorem 8 is that the global minimum of convex, non-negative and differentiable loss functions (including negative cross-entropy and squared error) is reached when the Jacobian becomes rank 1 and aligned with the training data.  A related theoretical result for linear networks is that the rank of the Jacobian decreases exponentially to 1 with the number of layers (Feng et al., 2022). Thus, Jacobian alignment can and should be considered the “ultimate destination” of deep learning.
>
> Our main theoretical result is supported by substantial empirical evidence in the literature demonstrating that DNs have an implicit bias towards very low-rank solutions (Le & Jegelka, 2022; Huh et al., 2023; Timor et al., 2023; Yunis et al., 2024b; Galanti et al., 2025).  For example, (Feng et al., 2022) shows that a wide range of pre-trained, state-of-the-art networks with thousands of classes feature Jacobians with a rank of around 100.
>
> Moreover, in response to your query, we computed the effective rank of the Jacobians in two SOTA DNs: the Swim-B transformer and ConvNext models trained on ImageNet (1000 classes). Using statistics terminology, we found that 90% of the explained variance in the Jacobians is captured by around 50 principal components, meaning that these Jacobians can be interpreted as having an effective rank of around 50. We maintain, and will investigate in future research, that further training (e.g., using GrokAlign) would reduce the effective rank even closer to our theoretical reachable limit of 1. We will report on the results of this empirical study in the revised paper.
>
> Feng, Ruili, et al. "Rank diminishing in deep neural networks." Advances in Neural Information Processing Systems 35 (2022): 33054-33065.

---

> > ### Comment · Reviewer_4vUT · 2025-11-25
> >
> > After reading the authors’ response, I will maintain my original rating, which leans slightly toward acceptance. The rebuttal addresses my main concerns about clarity and provides reasonable justification for the low-rank Jacobian assumptions, but this work is somewhat outside my core area of expertise, so I am not fully comfortable taking a very strong stance either for or against acceptance.

---

### Official Review · Reviewer_PBfN · 2025-11-04

**Soundness:** 2
**Presentation:** 2
**Contribution:** 2
**Rating:** 4
**Confidence:** 3

**Summary:**

The paper examines the problem of late generalisation (also known as grokking) and proposes a novel framework based on the input-output Jacobian to investigate its dynamics. Specifically, the authors present a theoretical result indicating that during training, two distinct phases occur: the phase of centroid alignment, followed by the phase of Jacobian alignment responsible for increased robustness of the solution. The theoretical findings are backed by the empirical results and later set the stage for designing a regularisation (GrokAlign), which is designed to induce the grokking faster.

**Strengths:**

1. The topic of NNs generalisation and its mechanistic understanding is a timely and important topic for the whole community.

2. The presentation of the core results is clear and relatively easy to follow, although the work is based on the previous works whose results are important to understand the whole framework in detail.

**Weaknesses:**

Clarity:
1. I'm still not sure about the link between Theorem 4 (the formation of centriod alignment) and the generalization. In Section 2.2. the authors only vaguely describe this relationship. Except for the authors' words, I don't find any experimental result supporting this relationship. Could the authors please elaborate on this more broadly?

Novelty & Significance:

1. The theoretical framework seems to hinge on the assumption that the input-output Jacobian converges to rank-1 matrix. Clearly, this is not the case in most of the situations (as even the authors note). Thus, it raises a question of the significance of the theoretical results when we know upfront they do not apply to real-world scenarios. Is there something I'm missing here?

2. The authors propose a regularisation method based on the theoretical analysis of the input-ouptut Jacobian (GrokAlign). However, in the text, the authors have already noticed that "similar forms of Jacobian regularisation have been demonstrated to improve the robustness of DNs in prior work". What is the contribution of the GrokAlign method and how it differs from the already established methods?


The last thing that I'd like the authors to confront with is the Fit&Align phenomenon studied in several works. In particular [1] introduced the concept of "extremal vectors" which describe the direction of the weights during the training and observed that initially the weights align with the data and only after the initial alignment, the weights starts increasing in norm. I'm not implying here that this observation dimnish the significance of this paper's contribution, but I feel that it's important to discuss this work as a related observation.

[1] https://arxiv.org/abs/1803.08367

**Questions:**

I have posted all of my questions in the previous section.

---

> ### Author Response · Authors · 2025-11-20
>
> Thank you for your careful review of our paper and for pointing out that it is a timely contribution on an important topic for the entire ML community. In our updated version, we have addressed your concerns to clarify our contributions.

---

> ### Author Response · Authors · 2025-11-20
> **Concerns Regarding Connection to Generalisation**
>
> **“I'm still not sure about the link between Theorem 4 (the formation of centriod alignment) and the generalization.”**
>
> We will make this link clearer and more rigorous in our updated paper. We have both theoretical and empirical rationales for this link. Theoretically, in our submitted paper, we have analyzed a two-layer, scalar-output deep network (DN) and rigorously proved that a changing alignment between the centroids and the training data corresponds to changing values in the neural tangent kernel evaluated at the training data. Prior work (Woodworth et al., 2020; Moroshko et al., 2020) has connected the period of training where the neural tangent kernel changes to the feature learning regime of training. Additional prior work (Baratin et al., 2020, Fort et al., 2020, Paccolat et al., 2020) has empirically connected the feature learning regime of training to improved generalization. Furthermore, prior work cited in our submitted paper (Lyu et al., 2024; Rubin et al., 2024; Kumar et al., 2024) makes the connection between the feature learning regime and delayed generalization (aka, grokking). These papers show empirically that DNs that have commenced feature learning exhibit greater generalisation. We plan to update our paper with the arguments above, and so would appreciate your comments. Nevertheless, in the absence of a direct rigorous theoretical result connecting centroid alignment to generalisation, in the updated paper, we will tone down our claims appropriately.
>
> Baratin, Aristide, et al. "Implicit regularization via neural feature alignment." International Conference on Artificial Intelligence and Statistics. PMLR, 2021.
>
> Fort, Stanislav, et al. "Deep learning versus kernel learning: an empirical study of loss landscape geometry and the time evolution of the neural tangent kernel." Advances in Neural Information Processing Systems 33 (2020): 5850-5861.
>
> Paccolat, Jonas, et al. "Geometric compression of invariant manifolds in neural networks." Journal of Statistical Mechanics: Theory and Experiment 2021.4 (2021): 044001.
>
> **“I don't find any experimental result supporting this relationship”**
>
> Figure 4 in our submitted paper showed that, for a three hidden-layer, fully connected DN trained on MNIST, as centroid alignment increases, the test accuracy also increases. In our updated paper, we plan to update Figure 4 to directly show the positive correlation between centroid alignment and test accuracy by plotting their values as a scatter plot. This figure will provide even stronger empirical support for our claim that when a DN is becoming centroid aligned, it starts to exhibit generalisation.

---

> ### Author Response · Authors · 2025-11-20
> **Concern Regarding Rank-One Assumption**
>
> **“The theoretical framework seems to hinge on the assumption that the input-output Jacobian converges to rank-1 matrix. Clearly, this is not the case in most of the situations (as even the authors note). Thus, it raises a question of the significance of the theoretical results when we know upfront they do not apply to real-world scenarios. Is there something I'm missing here?”**
>
> Our main theoretical result in Theorem 8 is that the global minimum of convex, non-negative and differentiable loss functions (including negative cross-entropy and squared error) is reached when the Jacobian becomes rank 1 and aligned with the training data.  A related theoretical result for linear networks is that the rank of the Jacobian decreases exponentially to 1 with the number of layers (Feng et al., 2022). Thus, Jacobian alignment can and should be considered the “ultimate destination” of deep learning.
>
> Our main theoretical result is supported by substantial empirical evidence in the literature demonstrating that DNs have an implicit bias towards very low-rank solutions (Le & Jegelka, 2022; Huh et al., 2023; Timor et al., 2023; Yunis et al., 2024b; Galanti et al., 2025).  For example, (Feng et al., 2022) shows that a wide range of pre-trained, state-of-the-art networks with thousands of classes feature Jacobians with a rank of around 100.
>
> Moreover, in response to your query, we computed the effective rank of the Jacobians in two SOTA DNs: the Swim-B transformer and ConvNext models trained on ImageNet (1000 classes). Using statistics terminology, we found that 90% of the explained variance in the Jacobians is captured by around 50 principal components, meaning that these Jacobians can be interpreted as having an effective rank of around 50. We maintain, and will investigate in future research, that further training (e.g., using GrokAlign) would reduce the effective rank even closer to our theoretical reachable limit of 1. We will report on the results of this empirical study in the revised paper.
>
> Feng, Ruili, et al. "Rank diminishing in deep neural networks." Advances in Neural Information Processing Systems 35 (2022): 33054-33065.

---

> ### Author Response · Authors · 2025-11-20
> **Concern Regarding Novelty of GrokAlign**
>
> **“The authors propose a regularisation method based on the theoretical analysis of the input-ouptut Jacobian (GrokAlign). However, in the text, the authors have already noticed that "similar forms of Jacobian regularisation have been demonstrated to improve the robustness of DNs in prior work". What is the contribution of the GrokAlign method and how it differs from the already established methods?”**
>
> While various forms of Jacobian regularisation have been applied to DN learning in previous works, GrokAlign is distinctly different from earlier approaches and makes some significant theoretical and empirical advances. We will add a discussion of the following to our updated paper.
>
> GrokAlign differs from earlier approaches fundamentally in its motivation, which is to induce a Jacobian-aligned DN with low-rank Jacobians.
>
> GrokAlign also makes three significant contributions:
>
> First, in Table 1 of our paper, we show that GrokAlign significantly accelerates the onset of generalisation in a DN, up to 20 times  faster than baseline methods. This is a completely novel application of Jacobian regularisation, and demonstrates how it can lead to much more computationally efficient learning methods, which are crucial as the global energy footprint of deep learning training skyrockets.
>
> Second, the GrokAlign penalty in Section 4.1 is flexible and can be applied to not just the entire DN but to specific sub-networks. Since the rank of the input-output mapping Jacobians is bounded above by the rank of the Jacobians of any sub-network (this follows from the basic linear algebra fact that rank(AB)≤min(rank(A), rank(B)), GrokAlign’s penalty term in the loss function can be calculated from the Jacobians of any sub-network to induce overall low-rank Jacobians. Empirically, we have found this flexibility to be very useful for both reducing the computational cost of computing the value of the penalty term every training iteration and for increasing the stability of training.  Indeed, in the experiment reported in Figure 5, we applied GrokAlign to only the last fully connected layers of the DN and not the early feature learning layers. In the updated version of the paper, we will ensure that this flexibility and new design tradeoff is made clear.
>
> Third, the flexibility of the GrokAlign penalty allows it to be used to compute the average Frobenius norm of two sub-networks, enabling the penalty to closely approximate the Nuclear norm of the Jacobian (Scarvelis and Solomon, 2024). While direct minimization of the rank is combinatorially complex, minimization of the Nuclear norm, which is the convex relaxation of the rank (Recht et al., 2010)  is combinatorially feasible, particularly so using the trick of (Scarvelis and Solomon, 2024).
>
> Scarvelis, Christopher, and Justin M. Solomon. "Nuclear norm regularization for deep learning." Advances in Neural Information Processing Systems 37 (2024)
>
> Guaranteed Minimum-Rank Solutions of Linear Matrix Equations via Nuclear Norm Minimization, Benjamin Recht, Maryam Fazel, and Pablo A. Parrilo, SIAM Review 2010 52:3, 471-501

---

> ### Author Response · Authors · 2025-11-20
> **Connection to Fit&Align**
>
> Thank you for pointing out the potential connections to Fit&Align, which bolsters the arguments we make in this paper. The conclusions of Fit&Align are similar to those in our paper in that the authors identify how the intrinsic characteristics of DNs align in specific directions.
>
> The first difference is that Fit&Align studies the alignment of weight vectors and we study the alignment of the centroids (which are a nontrivial function of the weight vectors). Fit&Align demonstrates that the weight vectors align in specific directions that are constructed based on the input points within “sectors”. In our paper, we show that centroids align in the direction of the input point at which they were computed. Since in ReLU DNs, the Jacobians are constructed by multiplying rows of the weight matrices based on the activation patterns of the input, there is likely an elegant connection between centroid alignment and Fit&Align. We will comment on this potential connection in the revised paper as an interesting direction for near future work.
> The second difference is that the analysis of Fit&Align is conducted only for one hidden layer ReLU networks, whereas our centroid analysis extends to arbitrary DNs.

---

### Meta-Review · Area_Chair_DuSS · 2025-12-27

**Summary:**

The submission studies grokking through the evolution of the input space tessellation induced by a network and the input–output sample Jacobian $J_x f$ at a point $x$. It introduces two alignment notions: centroid alignment, where $x$ (transposed) is colinear with the sum of the rows of $J_x f$, and Jacobian alignment, where $x$ is colinear with each row of $J_x f$. The paper develops this perspective theoretically in simplified settings and provides empirical evidence on standard architectures and datasets suggesting that these quantities correlate with phases of generalization and robustness. Motivated by this viewpoint, it proposes GrokAlign, a Jacobian-based regularization intended to encourage these alignments and to speed up delayed generalization while improving robustness in the reported experiments.


Reviewers generally find the paper well motivated and consider the overall approach interesting. At the same time, several reviewers raise substantive concerns about the strength and clarity of the central claims. In particular, the claimed connection between centroid alignment and generalization (e.g., as largely proxied by a change in the NTK in Theorem 4) is viewed as heuristic and insufficiently supported. Related concerns are raised about the reliance on (near) rank-one Jacobian assumptions. The empirical evidence is seen as suggestive but limited, with some reviewers judging the performance differences between GrokAlign and baseline methods to be small and insufficiently explored. While the rebuttal constructively addresses several points and outlines planned revisions, the theoretical gaps and empirical limitations appear too substantial to be fully resolved within a revision. Overall, the paper is regarded as offering a potentially valuable perspective, but with central implications that are currently insufficiently supported.

**Reviewer Concerns:**

Addressed: the rebuttal further motivates the rank-one/low-rank Jacobian premise (partially addressed), clarifies how GrokAlign is intended to differ from prior Jacobian regularization, and acknowledges the suggested connection to Fit&Align.

Outstanding: as indicated above in the meta-review.

**Reviewer Scores:**

All reviewers who objected to Theorem 4 are likely to maintain the view that the paper does not provide sufficient theoretical or empirical evidence that the proposed alignment phases meaningfully imply generalization or robustness. In particular, this applies to reviewers 3JCQ and Kxzz, with Kxzz explicitly noting in the rebuttal discussion that they would not revise their score. Reviewer 4vUT, who initially gave a score of 6, also indicated after the rebuttal that they were not comfortable taking a stronger stance in favor of acceptance.

---

### Decision · Program_Chairs · 2026-01-26

Reject